# Subspace Inference Enables Efficient Active Reward Learning from Preferences

## Abstract

Reinforcement learning from human feedback (RLHF) has emerged as a powerful yet sample-inefficient approach for learning reward models from human preferences, making active learning a critical component in synthesizing informative preferences queries. However, effective uncertainty quantification required for active learning remains a key challenge for large neural network reward models. In this paper, we introduce PreferenceEKF, a sample-efficient approach that tracks reward model uncertainty by framing active preference learning as a sequential Bayesian filtering problem. Instead of relying on computationally prohibitive posterior inference over the full neural network parameter space, our method performs sequential inference via extended Kalman filter within a low-dimensional parameter subspace, continuously updating reward model posterior as new preference queries arrive. Our approach enables scalable sampling of neural network parameters to efficiently compute acquisition functions for active reward learning. Experiments on the D4RL and V-D4RL benchmarks demonstrate that our approach achieves better sample efficiency, runtime, scalability, and calibration compared to other Bayesian deep learning approaches, and the learned reward models lead to competitive offline reinforcement learning policy performance. This highlights the potential of scalable Bayesian methods for preference-based reward modeling in RLHF. [1]

## 1 Introduction

In recent years, reinforcement learning from human feedback (RLHF) has become the dominant technique for aligning decision-making agents with human intentions (Christiano et al., 2017; Ouyang et al., 2022). The ease of providing preference feedback has been a crucial factor in their popularity as a feedback type for reward modeling, but since each feedback provides at most one bit of information, they are also known for their poor sample efficiency; asking a human thousands of comparison questions to learn a reward model (RM) is often not scalable.

A core problem of RLHF is active learning, where we want an agent to be judicious about the queries it asks to learn about a human's preferences as efficiently as possible (Sadigh et al., 2017; Casper et al., 2023; Baraka et al., 2025). Many active learning approaches require probabilistic modeling of uncertainty for computing data acquisition functions, making proper uncertainty representation an active area of research (Ovadia et al., 2019; Tran et al., 2020; Papamarkou et al., 2024). While Bayesian methods are well-principled, they are hard to scale to large neural networks (NN) (Izmailov et al., 2021). On the other hand, the simplicity of ensemble method (Dietterich, 2000; Lakshminarayanan et al., 2016) and dropout (Srivastava et al., 2014; Gal & Ghahramani, 2016) has made them popular choices for active learning. However, training multiple models can be computationally intensive, especially for large NN reward models. Although dropout gets around this issue, its effectiveness has been a subject of debate (Fort et al., 2020; Osband et al., 2022; Hron et al., 2018).

Due to recent advancements in approximate inference, Bayesian deep learning has become increasingly scalable (Daxberger et al., 2024; Shen et al., 2024). In this work, we develop a method called PreferenceEKF that enables efficient training of Bayesian neural networks for representing reward models in active preference-based reward learning. Specifically, by performing Bayesian filtering in a constructed neural network subspace, we

---

[1] Our code is anonymously available at `https://github.com/preferenceEKF2025/preference_ekf`.

maintain model uncertainty in a compute- and memory-efficient manner. The reduced dimensionality of the subspace enables application of the extended Kalman filter (EKF), a classic inference method, for training neural networks. This allows sampling of arbitrary number of reward models from the model posterior, and usage of the samples for computing common uncertainty-based acquisition functions such as expected information gain and disagreement (Hennig & Schuler, 2012; Hernández-Lobato et al., 2014; Bıyık et al., 2022).

To the best of our knowledge, we are the first to leverage subspace filtering to train neural network reward models from preference feedback. We compare our method, PreferenceEKF, to four widely used Bayesian deep learning methods for active preference-based reward learning in continuous control tasks. We further test whether the learned reward models are useful for policy optimization in offline RL tasks (Levine et al., 2020). We conduct our experiments in the D4RL (Fu et al., 2020) and the V-D4RL (Lu et al., 2023) benchmarks. Our findings are as follows:

- PreferenceEKF performs on par with or better than all Bayesian deep learning baselines in terms of sample efficiency and calibration in preference modeling tasks.
- PreferenceEKF's runtime is not only much faster, but also scales better with both model size and number of posterior samples compared to all other methods.
- When used for policy optimization, reward models learned using PreferenceEKF resulted in policies with performance competitive with those optimized with reward models learned using other methods.

## 2   Related Work

**Reinforcement learning from human preferences.** While early works in reward learning focused on learning from expert demonstrations (Abbeel & Ng, 2004; Finn et al., 2016; Ho & Ermon, 2016), much of recent interest have focused on reward learning from pairwise comparisons where human annotators are asked to compare two potential outcomes, e.g., labels, responses, or trajectories (Wirth et al., 2017; Christiano et al., 2017; Brown et al., 2019). Although preference feedback is much easier for annotators to provide than demonstrations, the minimal amount of information contained within a binary preference query necessitates collection of large amounts of feedback data.

Active learning is a widely used approach for minimizing the time-consuming process of collecting human feedback. It is a sequential problem in nature, as it iteratively collects the most useful data sample based on the model's current state, such as parameter posterior uncertainty. (Sadigh et al., 2017; Settles, 2009). While Bayesian methods have been successfully applied to obtain posteriors for active reward learning using lower-dimensional linear and Gaussian process reward models (Bıyık et al., 2022; 2024), it has not been widely adopted for neural reward models, since acquisition functions typically require sampling from the high-dimensional distribution of model parameters. Instead, ensembles and dropout have been the key enabler of neural network based active reward learning (Lee et al., 2021b; Christiano et al., 2017). Our work focuses on efficient yet performant posterior inference for active reward learning, without expensive training of multiple independent models.

**Uncertainty quantification for neural networks.** Classic Bayesian methods that have been successfully used for neural network uncertainty quantification include Laplace approximation (Daxberger et al., 2024), Hamiltonian Monte Carlo (Neal, 2011), and variational inference (Blei et al., 2017). While not strictly motivated by Bayesian principles, the simplicity of ensembling (Dietterich, 2000; Lakshminarayanan et al., 2016) and dropout (Srivastava et al., 2014; Gal & Ghahramani, 2016) has made them popular for uncertainty quantification (UQ). Although the dropout method gets around ensemble method's cost of training multiple independent models, it has been shown to lead to poor posterior approximation quality (Hron et al., 2018; Osband et al., 2022).

Bayesian filtering methods, which focuses on inferring hidden states from noisy observations, provide a principled approach to sequential learning, and have been widely used in robotics and signal processing (Thrun et al., 2005; Särkkä & Svensson, 2023). Application of Bayesian filtering for training neural networks (Singhal & Wu, 1988; de Freitas et al., 2000) has only recently been applied to deep networks via subspace methods by Duran-Martin et al. (2022).

Instead of deriving epistemic uncertainty from posterior inference, a separate line of work has focused on leveraging nonparametric statistics techniques such as the bootstrap to perform UQ (Efron, 1992), and has successfully applied this technique for exploration in deep reinforcement learning (Osband et al., 2018; 2016). The same group of authors have also leveraged joint predictions for UQ, and has applied the idea to finetuning large language models (Osband et al., 2023b;a). Our work leverages Bayesian filtering to train neural network reward models in active reward learning settings, where we focus primarily on parameter uncertainty instead of joint predictive uncertainty.

**Subspace methods for neural networks.** While there exists a vast literature on decreasing neural network size for efficient training and serving via architecture search (Elsken et al., 2019), quantization (Gholami et al., 2021), and pruning (Frankle & Carbin, 2022), we focus only on works that enable tractable inference in the reduced model. Specifically, there is growing evidence that the number of parameters required for a neural network to solve a task is often much smaller than the network's total parameter count; optimization and inference in the subspace spanned by these sub-parameters offer not only computational efficiency, but also tractability of applying Bayesian methods for neural network training (Fort et al., 2020; Larsen et al., 2022). These parameters are found either as a subset of neural network parameters, or within a lower-dimensional subspace thereof.

Methods focusing on parameter subsets typically apply Bayesian methods such as Bayesian linear regression or variational inference to the last layer of the neural network, and point estimation methods like stochastic gradient descent (SGD) to the intermediate layers (Snoek et al., 2015; Harrison et al., 2023; Brunzema et al., 2024). On the other hand, subspace methods typically construct the low-dimensional subspaces via either random projection or singular value decomposition of SGD iterates of the full network; any inference or optimization technique such as sliced sampling (Izmailov et al., 2020) or SGD Li et al. (2018) can then be applied in the subspace in a tractable manner.

## 3 Preliminaries

**Preference-based reward modeling.** We consider a Markov decision process (MDP) $\langle \mathcal{S}, \mathcal{A}, \mathcal{T}, r, \gamma \rangle$ with state space $\mathcal{S}$, action space $\mathcal{A}$, transition function $\mathcal{T}$, reward function $r : \mathcal{S} \to \mathbb{R}$, and discount factor $\gamma \in [0, 1)$. We assume access to a dataset of trajectories $\mathcal{D}^{traj} = \{\tau_1, \ldots, \tau_N\}$, where each trajectory $\tau_i$ is a sequence of $T$ steps $\tau_i = \{(s_{i,t}, a_{i,t}, s_{i,t+1})\}_{t=0}^{T-1}$, with each step consisting of state $s_t \in \mathcal{S}$, action $a_t \in \mathcal{A}$, and next-state $s_{t+1} \in \mathcal{S}$. In preference-based reward modeling, we do not assume access to a reward function. Instead, our task supervision comes from annotators who provide binary preference labels over pairwise trajectory comparisons, and the goal is to learn the annotator's reward function that informed their preference.

Formally, an annotator takes a query $Q_i = (\tau_a^i, \tau_b^i)$, and returns a preference label over the two trajectories $y_i = \mathbb{1}(\tau_a^i \succ \tau_b^i) \in \{0, 1\}$ according to their internal reward function $r$. Given a dataset of queries and responses $\mathcal{D} = \{Q_i, y_i\}_i$, a widely-used approach for preference learning is to approximate $r$ with a parameterized reward model $r_{\boldsymbol{\theta}}$ via maximum likelihood estimation, where the likelihood $p_\theta(y \mid \tau_a, \tau_b)$ is typically defined using the Bradley-Terry (BT) model (Bradley & Terry, 1952),

$$
\begin{aligned}
p_\theta(y \mid \tau_a, \tau_b) &= p_\theta(\tau_a \succ \tau_b) \\
&= \frac{\exp(\beta \cdot \mathcal{R}_{\boldsymbol{\theta}}(\tau_a))}{\exp(\beta \cdot \mathcal{R}_{\boldsymbol{\theta}}(\tau_a)) + \exp(\beta \cdot \mathcal{R}_{\boldsymbol{\theta}}(\tau_b))} \,.
\end{aligned}
\tag{1}
$$

In particular, $\beta$ is a temperature parameter that models noisily optimal behavior of an annotator, and $\mathcal{R}_{\boldsymbol{\theta}}(\tau_i)$ is the return of trajectory $\tau_i$ where the per-timestep reward is computed using a neural network-based RM $r_{\boldsymbol{\theta}}$, i.e., $\mathcal{R}_{\boldsymbol{\theta}}(\tau_i) = \sum_{t=0}^{T-1} r_{\boldsymbol{\theta}}(s_{i,t})$ (Lee et al., 2021a). [2]

**Information-theoretic active learning.** We adopt the acquisition function from Bıyık et al. (2020), InfoGain, for active preference-based reward learning, which assumes a distribution over RM parameters $p(\boldsymbol{\theta})$ such that, given a query-response pair $(Q_i, y_i)$ the predictive distribution is given by $p(y \mid Q) = \mathbb{E}_{p(\boldsymbol{\theta})}[p(y \mid$

---

[2]This formalism extends to state or state-action RMs, and whole trajectories or partial trajectory segments. Our experiments use state-based RM and partial trajectories.

$Q, \boldsymbol{\theta}$]. Using InfoGain leads to selection of the query $Q_i$ that maximizes the mutual information between the query's preference label $y_i$ and model parameter $\boldsymbol{\theta}$:

$$Q_i^* = \underset{Q_i}{\arg\max}\ I\left(\boldsymbol{\theta}; y_i \mid Q_i, \boldsymbol{b}^{i-1}\right) \tag{2a}$$

$$= \underset{Q_i}{\arg\max}\ H\left(y_i \mid Q_i, \boldsymbol{b}^{i-1}\right) - \mathbb{E}_{\boldsymbol{\theta}}\left[H(y_i \mid \boldsymbol{\theta}, Q_i)\right] \tag{2b}$$

where $I$ is the mutual information, $H$ is the Shannon entropy (Cover & Thomas, 2006), and $\boldsymbol{b}^{i-1} = p(\boldsymbol{\theta} \mid \mathcal{D}_{1:i-1})$ is the posterior distribution over RM parameters after learning from $(i-1)$ queries. This acquisition function is developed from the Bayesian active learning literature, which we detail in Section A.2.5. We approximate this acquisition function via sampling as follows:

$$Q_i^* \doteq \underset{Q_i}{\arg\max}\ \frac{1}{M} \sum_{y_i \in \{0,1\}} \sum_{\boldsymbol{\theta} \in \boldsymbol{\Theta}} P(y_i \mid Q_i, \boldsymbol{\theta}) \log_2\left(\frac{M \cdot P\left(y_i \mid Q_i, \boldsymbol{\theta}\right)}{\sum_{\boldsymbol{\theta}' \in \boldsymbol{\Theta}} P\left(y_i \mid Q_i, \boldsymbol{\theta}'\right)}\right) \tag{3}$$

where $\boldsymbol{\Theta}$ is the set of models sampled from the posterior $\boldsymbol{b}^{i-1}$, and $M$ is the number of drawn samples. This approximation is asymptotically equivalent to Eq. 2b as $M \to \infty$. We refer to Section 9.1 of Bıyık et al. (2020) for the proof. Due to the necessity of sampling models from the model posterior $\boldsymbol{b}^{i-1}$, Bıyık et al. (2020) restricted themselves to low-dimensional RMs, such as linear models. We now present our method, PreferenceEKF, which enables sampling of high-dimensional RMs like neural networks, that in turn allows us to scalably compute sampling-based acquisition functions like InfoGain to perform active learning.

## 4 Method

Sampling neural network models to approximate acquisition functions as in Eq. 3 can be expensive due to the high-dimensional parameter space of neural networks (Izmailov et al., 2021). We leverage the insight that neural networks are overparameterized and that solutions actually live in a much smaller subspace (Li et al., 2018), and perform posterior inference within this subspace. This allows us to sample an arbitrary number of models from a lower-dimensional posterior to approximate Eq. 3, without, e.g., the overhead of training ensembles. We first show how to use extended Kalman filter (EKF), a widely used filtering algorithm, to train neural network reward models from preference data, then we show how to scale EKF to deep neural networks using subspace methods (Duran-Martin et al., 2022). The full algorithm is shown in Algorithm 1.

**EKF for training neural networks.** Using the formulation of sequential Bayesian inference, we perform posterior inference of neural network parameters from streaming data $\mathcal{D}_{1:i-1} = \{(Q_1, y_1), \ldots, (Q_{i-1}, y_{i-1})\}$. Starting from some prior belief $\boldsymbol{b}^0 = p(\boldsymbol{\theta})$ on the parameters, our posterior after observing $i$ samples can be expressed using Bayes' rule as follows:

$$p(\boldsymbol{\theta}_i \mid \mathcal{D}_{1:i}) \propto \underbrace{p(\mathcal{D}_i \mid \boldsymbol{\theta}_i)}_{\text{Measurement}}\ p(\boldsymbol{\theta}_i \mid \mathcal{D}_{1:i-1})$$

$$p(\boldsymbol{\theta}_i \mid \mathcal{D}_{1:i-1}) = \int \underbrace{p(\boldsymbol{\theta}_i \mid \boldsymbol{\theta}_{i-1})}_{\text{Dynamics}}\ \underbrace{p(\boldsymbol{\theta}_{i-1} \mid \mathcal{D}_{1:i-1})}_{\text{Previous posterior}} d\boldsymbol{\theta}_{i-1} \tag{4}$$

where $p(\boldsymbol{\theta}_{i-1} \mid \mathcal{D}_{1:i-1})$ is the posterior belief over parameters after observing $i-1$ samples, which is combined with a parameter dynamics model and measurement model to form the posterior after observing the $i^{\text{th}}$ example. This formulation naturally allows for recursive estimation of model parameters by observing samples one at a time.

To tractably compute Eq. 4, we assume additive Gaussian noise for the dynamics model $p(\boldsymbol{\theta}_i \mid \boldsymbol{\theta}_{i-1}) = \mathcal{N}(\boldsymbol{\theta}_i \mid g(\boldsymbol{\theta}_{i-1}), \mathbf{U})$ and the measurement model $p(\mathcal{D}_i \mid \boldsymbol{\theta}_i) = \mathcal{N}(y_i \mid h(\boldsymbol{\theta}_i, Q_i), \mathbf{V})$, where $\mathbf{U} \in \mathbb{R}^{|\boldsymbol{\theta}| \times |\boldsymbol{\theta}|}$ and $\mathbf{V} \in \mathbb{R}^{|y| \times |y|}$ are prespecified Gaussian noise covariance. We treat neural network parameters as hidden states, and model the state dynamics $g(\boldsymbol{\theta}_{i-1}) : \mathbb{R}^{|\boldsymbol{\theta}|} \to \mathbb{R}^{|\boldsymbol{\theta}|}$ using an identity function. For preference learning, we model measurements $h(\boldsymbol{\theta}_i, Q_i) : \mathbb{R}^{|\boldsymbol{\theta}|} \times \mathbb{R}^{|Q|} \to \mathbb{R}^{|y|}$ using BT model $p_{\boldsymbol{\theta}}(\tau_a \succ \tau_b)$ computed using the

learned RM $r_\theta$ (Eq. 1). These assumptions make the model parameter inference objective in Eq. 4 solvable in closed-form with the EKF algorithm, where the posterior takes a Gaussian form $\boldsymbol{b}^i = \mathcal{N}(\boldsymbol{\mu}_i, \boldsymbol{\Sigma}_i)$ with $\boldsymbol{\mu}_i \in \mathbb{R}^{|\boldsymbol{\theta}|}$ and $\boldsymbol{\Sigma}_i \in \mathbb{R}^{|\boldsymbol{\theta}| \times |\boldsymbol{\theta}|}$. In Section A.1, we show the exact form of the EKF update procedure and provide further discussion on its linearized Gaussian form of the BT likelihood.

**Subspace inference.** Inference in the full parameter space of a neural network is difficult, as the size of the covariance matrix $\boldsymbol{\Sigma}_i$ of the Gaussian posterior scales in $O(|\boldsymbol{\theta}|^2)$. We instead perform EKF in a learned subspace of the NN: we denote the full space parameter as $\boldsymbol{\theta}$ and subspace parameter as $\boldsymbol{z}$, where $|\boldsymbol{z}| \ll |\boldsymbol{\theta}|$, resulting in posterior $\boldsymbol{b}^i = \mathcal{N}(\boldsymbol{\mu}'_i, \boldsymbol{\Sigma}'_i)$ where $\boldsymbol{\mu}'_i \in \mathbb{R}^{|\boldsymbol{z}|}$ and $\boldsymbol{\Sigma}'_i \in \mathbb{R}^{|\boldsymbol{z}| \times |\boldsymbol{z}|}$. We further assume a learned affine mapping $\boldsymbol{\theta}(\boldsymbol{z}) = \mathbf{A}\boldsymbol{z} + \boldsymbol{\theta}_*$ that allows us to transform the subspace parameters to the full space. Here $\boldsymbol{\theta}_*$ is initialized via SGD on a small warm-up dataset in the full space. $\mathbf{A} \in \mathbb{R}^{|\boldsymbol{\theta}| \times |\boldsymbol{z}|}$ is a fixed projection matrix obtained from applying SVD to the SGD iterates ran in the full space, as shown on Line 8 through Line 10. Alternatively, we can construct $\mathbf{A}$ via random projections without relying on SGD iterates. See Section A.2.3 for subspace construction details and Section 5.4 for an ablation study. We further note that although Algorithm 1 indicates reliance on an initial dataset, we show in Section A.2.8 that PreferenceEKF is still effective without it.

We perform EKF inference in the subspace to obtain an estimate $\boldsymbol{b}^i = p(\boldsymbol{z} \mid \mathcal{D}_{1:i})$ after observing each query-response pair $\mathcal{D}_i = (Q_i, y_i)$, then project each model sampled from $\boldsymbol{b}^i$ back to the full space via affine projection $\boldsymbol{\theta}(\boldsymbol{z})$ to perform the forward pass of the neural network to predict $\mathbb{1}(\tau_a^i \succ \tau_b^i)$. The predictive distribution is similarly computed via sampling followed by projection as $p(y \mid Q) = \mathbb{E}_{p(\boldsymbol{z})}[p(y \mid Q, \mathbf{A}\boldsymbol{z} + \boldsymbol{\theta}_*)]$. While computing the acquisition function and the predictive distribution require sampling from $\boldsymbol{b}^i$, the posterior update procedure itself is completely deterministic.

**Active learning using subspace inference.** We refer to our approach as PreferenceEKF, and draw comparison to the commonly used ensemble method, which we refer to as DeepEnsemble. We assume pool-based active learning (Settles, 2009), where we denote the pool of possible binary preference queries as $\mathcal{P}$.[3] For belief initialization (Line 12), whereas PreferenceEKF uses a zero-mean isotropic Gaussian of subspace dimension $|\boldsymbol{z}|$, DeepEnsemble initializes $M$ independent models each of dimension $|\boldsymbol{\theta}|$.

After belief initialization, the sequential phase of active learning begin. For random querying, Line 14 amounts to simply retrieving a random query from the query pool $\mathcal{P}$, whereas active learning algorithms computes an acquisition function for the optimal query to retrieve from the pool. The algorithm then receives the corresponding label for the retrieved query from an annotator in Line 15, where the annotator can either be a human-in-the-loop or a simulated oracle that expresses preference based on ground truth environment rewards. For belief update (Line 16), whereas PreferenceEKF performs Bayesian update in the constructed subspace on only the most recent query-response pair $\mathcal{D}_i$, DeepEnsemble trains each of the $M$ models using gradient descent on all data seen so far.

The most common uncertainty-based acquisition function is ensemble disagreement, i.e., pick the query $Q_i$ for which the predicted preference label $\mathbb{1}(\tau_a^i \succ \tau_b^i)$ has the highest variance across the ensemble. Disagreement has been popular for neural network-based active learning where it is expensive to scale Bayesian methods to high-dimensional settings (Christiano et al., 2017; Lee et al., 2021b), while InfoGain is the current state of the art acquisition function for lower-dimensional reward learning settings (Bıyık et al., 2020; 2024; Ellis et al., 2024). Although our method can be used to compute any sampling-based acquisition functions, we specifically leverage PreferenceEKF's ability to sample from high-dimensional distributions to scale InfoGain (Eq. 3) to neural network models. Due to the difficulty of sampling from high dimensional parameter distributions, DeepEnsemble approximates InfoGain by training multiple independent models, while Dropout does so by sampling parameter dropout masks during inference.

## 5 Experiments

**Baselines and Evaluation.** We compare PreferenceEKF to four Bayesian deep learning baselines commonly used for reward modeling: DeepEnsemble, Dropout, Laplace, and last-layer Markov chain Monte Carlo (LLMCMC), which we detail in Section A.2.4. We address the following questions: (1) Does preference-based

---

[3]Given a dataset of $N$ trajectories, there would be $|\mathcal{P}| = \binom{N}{2}$ possible pairwise comparison queries.

---

**Algorithm 1** PreferenceEKF for active preference-based reward learning

---

1:  **Input:**
2:  $\mathcal{P}$: Pool of all binary preference queries without labels
3:  $\mathcal{D}^{\text{init}} = \{(Q_i, y_i)\}_{i=1}^{\tau}$: Initial preference dataset with $\tau$ (query, label) pairs
4:  $B$: query budget limit
5:  $w$: number of SGD iterations for subspace construction
6:  **Procedure:**
7:  # Subspace Construction
8:  $\boldsymbol{\theta}_{1:w} = \text{SGD}(\mathcal{D}^{\text{init}})$                           $\triangleright$ $\boldsymbol{\theta}_{1:w} = [\boldsymbol{\theta}_1, \ldots, \boldsymbol{\theta}_w]$
9:  $\boldsymbol{\theta}_* = \boldsymbol{\theta}_w$                             $\triangleright$ Parameter offset: $\boldsymbol{\theta}_* \in \mathbb{R}^{|\boldsymbol{\theta}|}$
10: $\mathbf{A} = \text{SVD}(\boldsymbol{\theta}_{1:w})$                   $\triangleright$ Projection matrix: $\mathbf{A} \in \mathbb{R}^{|\boldsymbol{\theta}| \times |\boldsymbol{z}|}$
11: # Subspace Inference
12: $\boldsymbol{b}^0(\boldsymbol{z}) = \mathcal{N}(\boldsymbol{\mu}_0', \boldsymbol{\Sigma}_0')$
13: **for** $t = 1 : B$ **do**
14:     $Q_t = \text{ComputeQuery}(\boldsymbol{b}^{t-1}, \mathbf{A}, \boldsymbol{\theta}_*, \mathcal{P})$
15:     $y_t = \text{GetLabel}(Q_t)$
16:     $\boldsymbol{b}^t = \text{EKF}(\boldsymbol{b}^{t-1}, (Q_t, y_t))$
17: **end for**

---

reward learning with PreferenceEKF lead to better sample-efficiency, likelihood-based evaluation, and model calibration compared to the baselines? (2) Does subspace representation of parameter uncertainty $p(\boldsymbol{\theta} \mid \mathcal{D})$ lead to computational advantages over other representations such as ensembles and dropout masks? (3) How does the choice of subspace construction method impact our method's performance? (4) Can reward models sampled from PreferenceEKF's posterior be used for policy optimization via offline RL? (5) Can we extend our methods to image-based trajectories with sparse preference feedback?

In the reward learning experiments, given a limited query budget $B$, we would like to learn RMs from preference queries as sample-efficiently as possible. Evaluation is done by comparing the BT log-likelihood (Eq. 1) achieved by a RM on a held-out set of test queries throughout training. We focus our main experiments on synthetically generated preference labels. To create the preference query pool $\mathcal{P}$, we randomly sample pairwise partial trajectories from a trajectory dataset $\mathcal{D}^{traj}$, then generate noisily optimal synthetic labels as follows: for a given pair of trajectories, we compute their returns and sample a preference label according to the BT model, where a trajectory is preferred with probability exponentially proportional to its return.

In the offline RL experiments, the learned RMs are then used for training parameterized policies $\pi_\phi(a \mid s)$ via offline RL. This is done by first labeling the trajectory dataset $\mathcal{D}^{traj}$ with the learned RM: we take the average predicted reward over $M$ models $r_{\boldsymbol{\theta}}^M(s_{i,t}) = \frac{1}{M} \sum_{m=1}^{M} r_{\boldsymbol{\theta}}^m(s_{i,t})$ for each state, where $r_{\boldsymbol{\theta}}^m$ is the $m^{\text{th}}$ sampled reward model or ensemble member. A reward-labeled trajectory takes the form, $\tau_i = \{(s_{i,t}, a_{i,t}, s_{i,t+1}, r_{\boldsymbol{\theta}}^M(s_{i,t}))\}_{t=0}^{T-1}$. We train policies on the reward-labeled $\mathcal{D}^{traj}$ using Implicit Q-Learning (IQL) (Kostrikov et al., 2021), an empirically successful offline RL algorithm. We evaluate policies by comparing their empirical rollout returns throughout RL training.

**Tasks.** We evaluate our approach in D4RL (Fu et al., 2020), a popular offline RL benchmark, and choose a mixture of environments spanning MuJoCo locomotion (Todorov et al., 2012), Adroit Shadow Hand (Rajeswaran et al., 2018), and Maze2D navigation. Within each environment, we choose trajectory datasets of varying characteristics: MuJoCo trajectories (HalfCheetah, Hopper, Walker2d) span a range of performance quality, Adroit trajectories (pen twirling) are generated by a human operator and a fine-tuned expert-level RL policy, and maze navigation trajectories are collected from policies executed in mazes of varying difficulty. We consider each dataset as a separate task, for a total of 12 tasks.

**Implementation Details.** Unless otherwise stated, all experiments are done on a single node with 8 NVIDIA RTX A6000 GPUs via sharding, query budget $B = 60$, and trajectory segments of length 50. On the belief update step (Line 16), PreferenceEKF learns from only the most recent query-label pair, while all baselines learn from all data seen so far. With the exception of the scaling experiments in Section 5.2 and the

ablation experiments in Section 5.4, all reward models are represented as multi-layer perceptrons (MLP) with two hidden layers of 64 units, using subspace dimensionality $|z| = 200$.

### 5.1 Does PreferenceEKF lead to sample-efficient active reward learning?

Given a fixed query budget per task, we evaluate each method over 5 seeds. We use state-based partial trajectories, and compute return of each trajectory as $\mathcal{R}_{\boldsymbol{\theta}}(\tau_i) = \sum_{t=1}^{T} r_{\boldsymbol{\theta}}(s_{i,t})$. We show in Figure 1a that aggregated over all tasks (see Appendix A for details on task-aggregation), both random and active variants of PreferenceEKF performs on par with or outperforms all other baselines in terms of both sample efficiency and the final evaluated log-likelihood. We show per-task results in Figure A.2, where we found that in most tasks, active PreferenceEKF outperforms all other methods in terms of sample efficiency and final log-likelihood. We refer to Section A.2.1 for statistical significance tests backing up these empirical observations.

**On acquisition functions:** Although our main result was obtained with all methods using the InfoGain acquisition function, we show additional results using disagreement and entropy acquisition function in Section A.2.5. We found InfoGain to be the only acquisition function that enabled the active variant of all five methods to perform better than or on par with their random counterpart. This reflects findings from previous works that demonstrated InfoGain's higher sample-efficiency compared to other acquisition functions (Bıyık et al., 2020).

**On posterior sampling:** DeepEnsemble is the only method that needs to train multiple models to represent the posterior, so we set $M = 5$ as is commonly done for ensemble-based uncertainty quantification (Ovadia et al., 2019); all other methods can sample arbitrary number $M$ of models from the learned posterior, so we set $M = 100$ for them. This raises the question of whether our method's higher sample efficiency in preference learning is solely due to the larger number of posterior samples, or whether the learned posterior indeed captures the annotator's preference. For fair comparison, we show in Section A.2.6 results where all methods use the same number of models $M = 5$, and found PreferenceEKF to still be the most successful method in terms of sample-efficiency and final log-likelihood.

**On the unimodality of EKF:** Due to the unimodal Gaussian assumption that EKF places on the measurement function, dynamics functions, and the resulting posterior, we note that our approach is designed primarily for learning the preferences of only a single annotator. We show in Section A.2.9 results where we learn from crowd-sourced human annotators labels. Due to the multimodality of the resulting preference distribution, none of the methods achieved great log-likelihood evaluation. While we acknowledge pluralistic alignment as an critical open problem of RL from human feedback (Casper et al., 2023; Sorensen et al., 2024), we view our work as addressing the complementary and largely orthogonal challenge of improving sample efficiency of preference-based reward learning. Extending our framework to accommodate multimodal preference distributions (Myers et al., 2021) may be done by performing inference using non-parametric methods such as particle filters (Thrun et al., 2005), but this may come as a trade-off for inference efficiency in sequential learning settings. We leave such explorations to future work.

### 5.2 How does training runtime with PreferenceEKF scale?

We first note in Figure 1b that PreferenceEKF training is vastly faster than the baselines, achieving roughly $5\times$ speedup compared to DeepEnsemble and over $40\times$ speedup compared to LLMCMC. This is primarily due to the sequential nature of EKF, making it unnecessary for PreferenceEKF to repeatedly train / perform posterior updates on queries it has already seen. All baseline methods require this so as to prevent catastrophic forgetting, thus slowing down their runtime. We show results that relax this assumption in Section A.2.7. Lastly, note that the long runtime of LLMCMC is due to the necessity for the MCMC chain to converge, which takes upwards of a thousand steps for every posterior update upon receiving a new query.

Next, we investigate whether subspace filtering can serve as a scalable alternative to gradient descent for preference learning, with respect to both larger reward models and more model samples. As such, we only compare our method to DeepEnsemble and Dropout, which are primarily based on SGD. We run all scaling experiments on CPUs as the larger models and ensemble sizes led to out-of-memory errors on GPUs. We show in Figure 2a that given a fixed architecture of a two-layer MLP with 64 units per layer, the runtime of

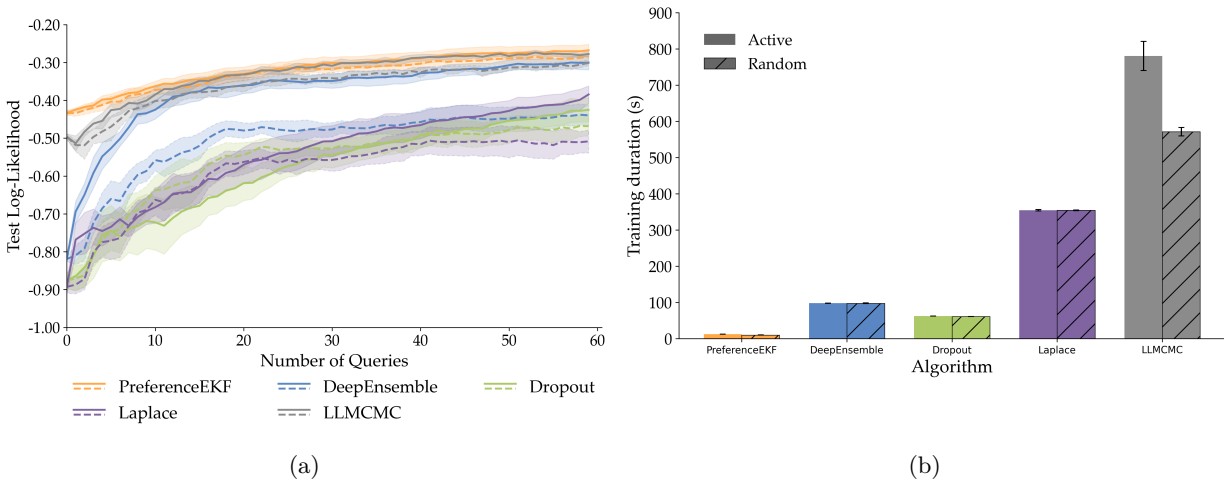

(a)                                                                    (b)

Figure 1: Figure 1a shows log-likelihood comparison of the random (dashed line) and active (solid line) variants of each method using the InfoGain acquisition function (higher means better fitting of annotator preference distribution). Figure 1b shows training runtime duration of both active and random variant of each method (lower means faster training). For table version of this plot, please see Section A.2.2. Each line plot and bar plot is aggregated over 12 D4RL tasks (mean $\pm$ 95% bootstrap confidence interval over 12 seeds)

PreferenceEKF for learning a reward model from $B = 60$ queries scales much more gracefully with increasing $M$ compared to other methods. While Dropout does not need to maintain multiple models, it is still slower than PreferenceEKF as it performs model update in full parameter space instead of a lower-dimensional subspace. Figure 2b demonstrates that final test log-likelihood favors PreferenceEKF over the other methods, showcasing that our approach maintains consistent performance on top of computational efficiency given increasing $M$. Figure 2c and Figure 2d show similar favorable scaling properties of PreferenceEKF except that we fix the number of model samples ($M = 5$) and increase the neural network architecture instead. This showcases the scalability of subspace training to not only settings where we need large number of model samples $M$, but also to settings where we need larger neural networks $|\boldsymbol{\theta}|$.

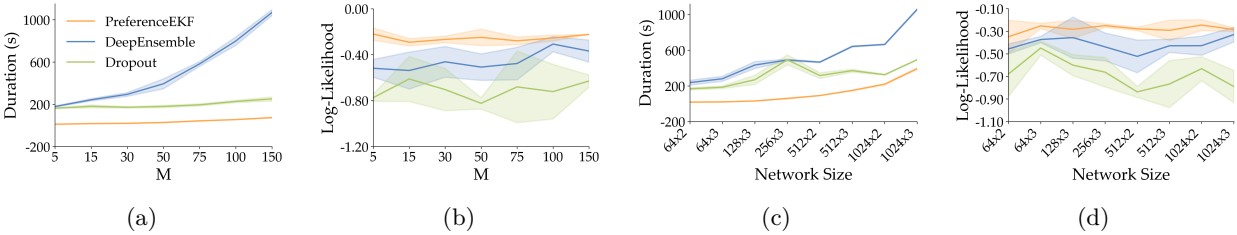

(a)                          (b)                          (c)                          (d)

Figure 2: Figure 2a and Figure 2b show how runtime scales with the number of model samples $M$ in the active learning setting (mean $\pm$ std over 3 seeds). Figure 2c and Figure 2d show runtime scaling with neural network architecture size. Overall, PreferenceEKF has the fastest runtime and the best scaling trend, while retaining high log-likelihood evaluation.

## 5.3 Does PreferenceEKF lead to better model calibration?

While effective representation of parameter uncertainty is crucial for efficient active learning, it is also important for calibration of model predictions (Guo et al., 2017; Ovadia et al., 2019). We study whether uncertainty quantification (UQ) using subspace inference methods leads to better calibrated model predictions compared to UQ using the baselines, as quantified by two commonly used UQ metrics: expected calibration error (ECE) (Naeini et al., 2015; Pavlovic, 2025) and Brier score (Brier, 1950; DeGroot & Fienberg, 1983).

We show in Figure 3a that PreferenceEKF has the lowest ECE among all methods, and the second lowest Brier score behind active DeepEnsemble. This highlights the quality of posterior approximation achieved by subspace inference methods compared to the other Bayesian deep learning baselines. We provide further calibration experiment details and reliability diagrams in Section A.2.12.

### 5.4 Ablation study on subspace construction

The method for subspace construction for PreferenceEKF can be modified to 1) use varying dimensionality of the subspace, and to 2) use random projection to generate the subspace basis instead of running SVD on SGD iterates (Li et al., 2018; Izmailov et al., 2020). While all of our experiments so far use a fixed dimensionality of $|\boldsymbol{z}| = 200$ with SVD-based construction, we perform an ablation analysis over these choices, as shown in Figure 3b. We observe that while the SVD-based approach works well for smaller subspace dimensions, the random projection approach can eventually reach performance on par with or even outperform the SVD approach as the subspace dimension increases.

We further show in Section A.2.8 that in the case where no initial dataset is available, belief initialization using the random projection approach is often sufficient good reward learning performance. This result decouples PreferenceEKF's reliance on SGD altogether. For consistency, unless otherwise stated, our main PreferenceEKF experiments are performed with the SVD-based approach that relies on SGD.

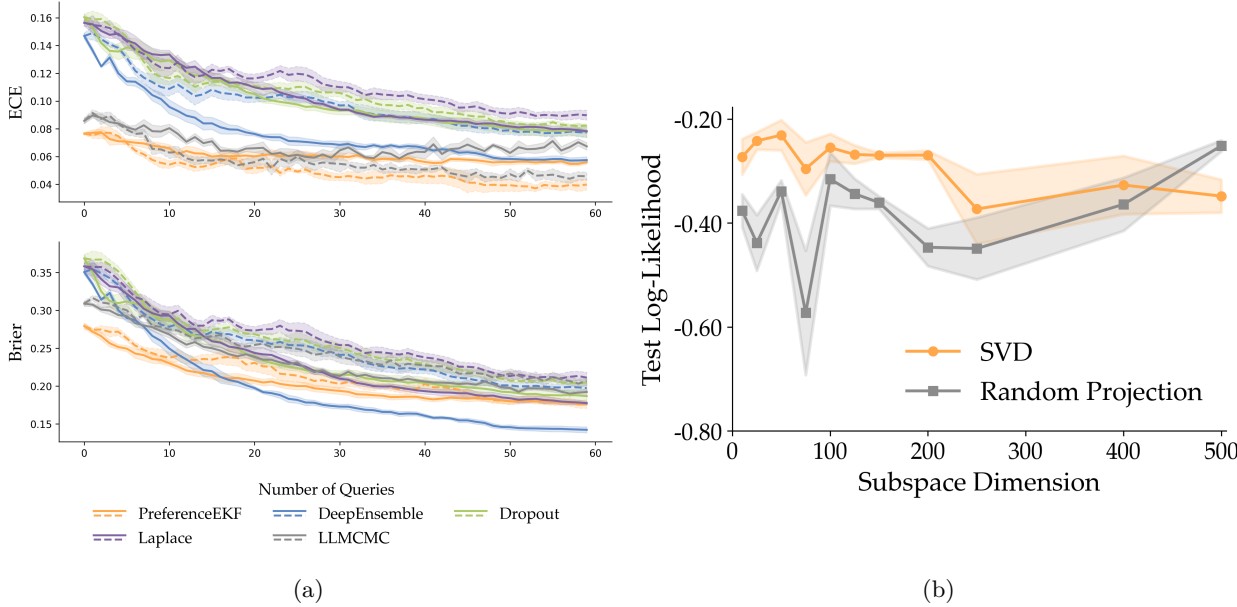

(a)                                                                 (b)

Figure 3: Figure 3a shows calibration results of the random (dashed line) and active (solid line) variants of the methods, as evaluated by expected calibration error (using 10 bins) and Brier score on a test dataset (lower is better for both metrics). Figure 3b shows an ablation over the subspace construction technique for PreferenceEKF, as evaluated by log-likelihood on a test dataset (higher is better). Both the UQ experiment and ablation analysis here are performed over 3 seeds (mean ± std) on the Walker Medium Expert task.

### 5.5 Can RMs learned using PreferenceEKF be used for policy optimization?

The goal of the offline RL experiments is to test whether a reward model learned from limited number of preference queries can be used to optimize a policy that reaches or exceeds the performance of a policy trained with ground-truth environment rewards (GT policy). All policies are trained using IQL (Kostrikov et al., 2021) over 5 seeds on the reward-labeled dataset for 1M steps, and evaluation is done via 5 rollouts every 50K steps. We show in Figure A.14 that aggregated across all tasks, policies induced by reward models learned from all active preference learning methods converge to similar policy performance, with all policies

performing on par with or slightly worse than the GT policy. This showcases that our method is capable of producing reward models suitable for policy optimization. As the primary goal of our work is to improve the sample efficiency of preference-based reward learning, we leave studies on the interplay between reward learning and policy learning to future work. We provide further discussion of this result in Section A.3.1.

### 5.6 Can PreferenceEKF learn from image data and sparse preference feedback?

While our main experiments showcase the effectiveness of PreferenceEKF in state-based control tasks, where preference labels are synthetically generated by comparing sum of dense rewards between two trajectories, we apply our method to two additional challenging yet common settings: (1) sparse comparative feedback, and (2) pixel-based control.

Our experiments thus far have relied on using dense trajectory rewards to generate synthetic preference labels, which has allowed us to perform preference learning on partial trajectory segments, thus easing the reward credit assignment problem (Wirth et al., 2017). However, real robot datasets often only have sparse binary success / failure labels for each trajectory, making it impossible to rely on dense comparative feedback signal for preference learning. We apply our method to this challenging setting, where we use full real robot trajectories from the SOAR dataset (Zhou et al., 2024) across multiple manipulation tasks, and observed favorable results for our approach, which we detail in Section A.2.10.

Despite subspace filtering's effectiveness in handling large parameter counts, the difficulty of scaling PreferenceEKF to pixel-based reward models is that the update step of EKF scales cubically with dimensionality of the observation space, which poses scalability challenges to high-dimensional inputs such as images. We resolve this issue by relying on pretrained image embeddings rather than raw pixel inputs, and observed promising results of active preference-based learning of pixel reward models. We refer to Section A.2.11 for results and further details on pixel-based tasks. Overall, we believe that the two favorable sets of results here highlight the applicability of our method to the high-dimensional and sparse feedback nature of common real robot data.

## 6 Conclusion

In this work, we successfully adopted extended Kalman filters to train neural networks in active preference-based reward modeling setting. We showed several advantages of maintaining a subspace distribution over neural network parameters $p(\boldsymbol{\theta} \mid \mathcal{D})$, in comparison to four other widely used Bayesian deep learning methods for active reward learning. Our approach led to more sample efficient active reward learning, similarly performant RL policy optimization, better runtime scaling with respect to model size and model sample count, and better calibration through higher-quality uncertainty representation.

**Limitations and future work.** While we found subspace method to be an effective tool for scaling Bayesian filtering methods for neural network training, it is unsure whether this approach will be effective for applying Bayesian methods to foundation model-scale reward models (Mahan et al., 2024; Zhang et al., 2024). Due to the unimodality of the Gaussian distribution that extended Kalman filter maintains, alternative methods may need to be investigated for approximating multimodal posteriors, e.g., learning reward functions from annotators with differing preferences (Poddar et al., 2024; Siththaranjan et al., 2023). We would further like to evaluate uncertainty quantification using the recent works on epistemic neural networks (Osband et al., 2023b), which focuses on joint predictions uncertainty instead of marginal predictive distribution.

Our work primarily focuses on improving sample-efficiency of reward modeling in RLHF, but we would like to further investigate how learned posterior distribution of reward models can aid in RL policy's exploration and serve as a mechanism for mitigating reward hacking (Yang et al., 2024a; Gao et al., 2022; Hadfield-Menell et al., 2017). Finally, due to its sample-efficiency and adaptivity to non-stationary distributions, we believe the subspace filtering method to be a viable candidate for uncertainty quantification and large model finetuning in robot learning domains (Bellemare et al., 2017; Fridovich-Keil et al., 2020; Bobu et al., 2020).

## Broader Impacts

Our work presents an algorithm for active learning in preference-based reward modeling, enhancing the efficiency and accuracy of neural network training in applications requiring subjective human evaluations, such as natural language processing, personalized recommendations, and human-robot interaction. By optimizing data collection around uncertain or high-impact preferences, our approach can reduce labeling costs and improve model alignment with human intentions. However, it is possible that working in the reduced subspace and performing inference with extended Kalman filter may introduce suboptimalities in preference modeling such as bias amplification or neglect of minority preferences. To mitigate these risks, future research should investigate the robustness of PreferenceEKF and potential information loss caused by subspace reduction.

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

# A Technical Appendices and Supplementary Material

Our code is available in the JAX (Bradbury et al., 2018) framework at `https://github.com/preferenceEKF2025/preference_ekf`. For implementation of the reward learning algorithms, we use Dynamax (Linderman et al., 2025) for extended Kalman filtering (EKF), Laplax (Weber et al., 2025) for Laplace approximation, and Blackjax (Cabezas et al., 2024) for MCMC. For offline RL, we use Unifloral (Jackson et al., 2025) for implementation of implicit Q-learning (IQL). All statistical testing are done using SciPy (Virtanen et al., 2020). Unless stated otherwise, all experiments are done on a single node with 8 NVIDIA RTX A6000 GPUs via SLURM sharding.

For figures that aggregate across tasks and per-task seeds (e.g., Figure 1a, Figure 1b, Figure 3a, Figure A.14), we aggregate as follows: given a dependent variable per step, we pool at each step across 12 tasks $\times$ $n$ seeds per task, and plot the per-step mean performance over $12n$ runs along with either standard error or 95% bootstrap interval for the confidence bounds. In our main preference learning results in Figure 1a, the dependent variable is test-likelihood for preference learning after every step of acquired query label. We take similar approaches for our calibration results in Figure 3a, where the dependent variable per step is expected calibration error or Brier score. For policy learning results in Figure A.14, the steps are evaluation rollouts every 40K gradient updates, and the dependent variable is environment rollout return.

## A.1 EKF with Bradley-Terry Likelihood

Here we provide the exact form of the EKF belief update procedure for posterior inference upon receiving a new query, where we use the BT model for the measurement function. For more details, please see chapter 8.3 of Murphy (2023a).

For convenience, we first reproduce the general form of the EKF update procedure from Section 4. Using the formulation of sequential Bayesian inference, we perform posterior inference of neural network parameters from streaming data $\mathcal{D}_{1:i-1} = \{(Q_1, y_1), \ldots, (Q_{i-1}, y_{i-1})\}$, where $Q_i = \{\tau_a, \tau_b\}$ is the pairwise preference query and $y_i$ is the binary preference label. Starting from some prior belief $b^0 = p(\boldsymbol{\theta})$[4] on the parameters, our posterior after observing $i$ samples can be expressed using Bayes' rule as follows:

$$p(\boldsymbol{\theta}_i \mid \mathcal{D}_{1:i}) \propto \underbrace{p(\mathcal{D}_i \mid \boldsymbol{\theta}_i)}_{\text{Measurement}} \; p(\boldsymbol{\theta}_i \mid \mathcal{D}_{1:i-1})$$

$$p(\boldsymbol{\theta}_i \mid \mathcal{D}_{1:i-1}) = \int \underbrace{p(\boldsymbol{\theta}_i \mid \boldsymbol{\theta}_{i-1})}_{\text{Dynamics}} \underbrace{p(\boldsymbol{\theta}_{i-1} \mid \mathcal{D}_{1:i-1})}_{\text{Previous posterior}} d\boldsymbol{\theta}_{i-1} \tag{5}$$

where $p(\boldsymbol{\theta}_{i-1} \mid \mathcal{D}_{1:i-1})$ is the posterior belief over parameters after observing $i-1$ samples, which is combined with a parameter dynamics model and measurement model to form the posterior after observing the $i^{\text{th}}$ example $\mathcal{D}_i$. We assume additive Gaussian noise for both the dynamics model $p(\boldsymbol{\theta}_i \mid \boldsymbol{\theta}_{i-1}) = \mathcal{N}(\boldsymbol{\theta}_i \mid g(\boldsymbol{\theta}_{i-1}), \mathbf{U})$ and the measurement model $p(\mathcal{D}_i \mid \boldsymbol{\theta}_i) = \mathcal{N}(y_i \mid h(\boldsymbol{\theta}_i, Q_i), \mathbf{V})$, where $\mathbf{U} \in \mathbb{R}^{|\boldsymbol{\theta}| \times |\boldsymbol{\theta}|}$ and $\mathbf{V} \in \mathbb{R}^{|y| \times |y|}$ are prespecified Gaussian noise covariance matrices, and $g : \mathbb{R}^{|\boldsymbol{\theta}|} \to \mathbb{R}^{|\boldsymbol{\theta}|}$ and $h : \mathbb{R}^{|\boldsymbol{\theta}|} \times \mathbb{R}^{|Q|} \to \mathbb{R}^{|y|}$ are deterministic dynamics function (how neural network model parameters change over time) and measurement function (likelihood of observed preference data given current model parameters), respectively.

To apply the above formalism to preference learning of neural network reward model, we model the dynamics using an identity function $g(x) = x$, and the measurements using the BT model $h(\boldsymbol{\theta}_i, \mathcal{D}_i) = p_\theta(y \mid \tau_a, \tau_b) = p_\theta(\tau_a \succ \tau_b)$ computed using the learned RM $r_{\boldsymbol{\theta}}$ (Eq. 6):

$$p_\theta(y \mid \tau_a, \tau_b) = p_\theta(\tau_a \succ \tau_b)$$

$$= \frac{\exp(\beta \cdot \mathcal{R}_{\boldsymbol{\theta}}(\tau_a))}{\exp(\beta \cdot \mathcal{R}_{\boldsymbol{\theta}}(\tau_a)) + \exp(\beta \cdot \mathcal{R}_{\boldsymbol{\theta}}(\tau_b))} \; . \tag{6}$$

Assumptions on additive Gaussian noise and nonlinear dynamics and measurement functions make the neural network inference objective in Eq. 4 solvable in closed-form with the EKF algorithm, where the posterior

---

[4]We use $\boldsymbol{\theta}$ to broadly refer to model parameters, but for subspace inference specifically, our belief is over $\boldsymbol{z}$.

takes a Gaussian form $\boldsymbol{b}^i = p(\boldsymbol{\theta}_i \mid \mathcal{D}_{1:i}) = \mathcal{N}(\boldsymbol{\mu}_i, \boldsymbol{\Sigma}_i)$ with mean $\boldsymbol{\mu}_i \in \mathbb{R}^{|\boldsymbol{\theta}|}$ and covariance $\boldsymbol{\Sigma}_i \in \mathbb{R}^{|\boldsymbol{\theta}| \times |\boldsymbol{\theta}|}$. For belief initialization, we set $\boldsymbol{\mu}_0$ to be the zero vector and $\boldsymbol{\Sigma}_0$ to be a diagonal matrix.

The EKF algorithm alternates between a belief prediction step and a belief update step to update $\boldsymbol{b}^i = p(\boldsymbol{\theta}_i \mid \mathcal{D}_{1:i}) = \mathcal{N}(\boldsymbol{\mu}_i, \boldsymbol{\Sigma}_i)$ in light of new data $D_i = \{Q_i, y_i\}$. The predict step is as follows, using the identity function for model parameter dynamics function $g(x) = x$:

$$
\begin{aligned}
\boldsymbol{\mu}_{i|i-1} &= g\left(\boldsymbol{\mu}_{i-1}\right) \\
&= \boldsymbol{\mu}_{i-1} \\
\boldsymbol{\Sigma}_{i|i-1} &= \mathbf{G}_i \boldsymbol{\Sigma}_{i-1} \mathbf{G}_i^\top + \mathbf{U},
\end{aligned}
\tag{7}
$$

where $\mathbf{G}_i \in \mathbb{R}^{|\boldsymbol{\theta}| \times |\boldsymbol{\theta}|}$ is the Jacobian matrix of the model dynamics function. In the case of an identity function, $\mathbf{G}_i$ is just an identity matrix.

The update step is as follows, using the BT likelihood for measurement function $h(\boldsymbol{\theta}_i, \mathcal{D}_i) = p_\theta(y \mid \tau_a, \tau_b) = p_\theta(\tau_a \succ \tau_b)$:

$$
\begin{aligned}
\hat{\boldsymbol{y}}_i &= h\left(\boldsymbol{\mu}_{i|i-1}, D_i\right) \\
&= p_\theta(y \mid \tau_a, \tau_b) \\
\mathbf{S}_i &= \mathbf{H}_i \boldsymbol{\Sigma}_{i|i-1} \mathbf{H}_i^\top + \mathbf{V}_i \\
\mathbf{K}_i &= \boldsymbol{\Sigma}_{i|i-1} \mathbf{H}_i^\top \mathbf{S}_i^{-1} \\
\boldsymbol{\mu}_i &= \boldsymbol{\mu}_{i|i-1} + \mathbf{K}_i\left(\boldsymbol{y}_i - \hat{\boldsymbol{y}}_i\right) \\
\boldsymbol{\Sigma}_i &= \boldsymbol{\Sigma}_{i|i-1} - \mathbf{K}_i \mathbf{S}_i \mathbf{K}_i^\top,
\end{aligned}
\tag{8}
$$

where $\mathbf{H}_i \in \mathbb{R}^{|y| \times |\boldsymbol{\theta}|}$ is the Jacobian matrix of the measurement function. In the case of the BT likelihood, $|y| = 2$ as it is a Bernoulli probability distribution given the return of two trajectories. Each row of $\mathbf{H}_i$ is just the gradient of the probability of preferring the corresponding trajectory over the other with respect to the reward model parameters (or subspace dimension thereof). We obtain both Jacobian matrices via Jax's automatic differentiation capability using the Dynamax library Bradbury et al. (2018); Linderman et al. (2025).

### A.1.1 On linearization of the Bradley-Terry Likelihood:

First recall that we denote $h(\theta_i, Q_i = \{\tau_a, \tau_b\})$ as the EKF measurement function that predicts the probability of a preference label $y_i$ for the pairwise query $Q_i$ given current reward model (subspace) parameters $\theta_i$. We further note that the BT likelihood of preference $\tau_a \succ \tau_b$ can be defined using the sigmoid function $h(\boldsymbol{\theta}) = P_{\boldsymbol{\theta}}(\tau_a \succ \tau_b) = \sigma\left(r_{\boldsymbol{\theta}}(\tau_a) - r_{\boldsymbol{\theta}}(\tau_b)\right)$. To apply EKF, we linearize $h(\theta_i, Q_i)$ around the mean of the current model parameter estimate (which we assume is also Gaussian) $\boldsymbol{\mu}_{i|i-1}$, which we obtain from EKF's prediction step. We apply first-order Taylor expansion:

$$
h(\boldsymbol{\theta}) \approx h\left(\boldsymbol{\mu}_{i|i-1}\right) + \mathbf{H}_i\left(\boldsymbol{\theta} - \boldsymbol{\mu}_{i|i-1}\right)
$$

, where $\mathbf{H}_i \in \mathbb{R}^{|y| \times |\theta|}$ is the Jacobian matrix of the measurement function, which captures the sensitivity of the linearized BT measurement function with respect to the parameters $\boldsymbol{\theta}$. We derive the explicit form of $\mathbf{H_i}$ using the chain rule. Recall the derivative of the sigmoid function $\sigma'(x) = \sigma(x)(1 - \sigma(x))$, and letting $z = r_\theta(\tau_a) - r_\theta(\tau_b)$:

$$
\mathbf{H}_i = \frac{\partial \sigma(z)}{\partial \boldsymbol{\theta}} = \sigma'(z)\nabla_\theta z = \sigma(z)(1 - \sigma(z))\left(\nabla_{\boldsymbol{\theta}} r_{\boldsymbol{\theta}}(\tau_a) - \nabla_{\boldsymbol{\theta}} r_{\boldsymbol{\theta}}(\tau_b)\right)
$$

We can interpret the term $\sigma'(z)$ as a weighting coefficient for the difference in reward model gradient $\left(\nabla_{\boldsymbol{\theta}} r_{\boldsymbol{\theta}}(\tau_a) - \nabla_{\boldsymbol{\theta}} r_{\boldsymbol{\theta}}(\tau_b)\right)$. We note that $\sigma'(z)$ is maximized at $(\max_z \sigma'(z) = 0.25)$ when $z = r_\theta(\tau_a) - r_\theta(\tau_b) = 0$, i.e., when both pairwise comparison items have the same reward and thus high uncertainty under the BT likelihood as to which item is preferred. Conversely, $\lim_{|z| \to \infty} \sigma'(z) = 0$, i.e., when one item has much higher reward than the other and thus strong confidence / low uncertainty under the BT likelihood, the difference

in reward model gradient vanishes. In summary, under the linearized measurement model, high reward model uncertainty over the preference label leads to to higher value for $\mathbf{H_i}$ and thus stronger updates to model parameters, as captured by Kalman gain $\mathbf{K}_i = \mathbf{\Sigma}_{i|i-1}\mathbf{H}_i^\top \left(\mathbf{H}_i \mathbf{\Sigma}_{i|i-1}\mathbf{H}_i^\top + \mathbf{V}\right)^{-1}$. With low uncertainty and low $\mathbf{H_i}$, Kalman gain $\mathbf{K_i}$ tends towards zero, making small or no updates to model parameters.

### A.1.2 On the locally Gaussian assumption of the Bradley-Terry Likelihood:

BT distribution is inherently a Bernoulli distribution, which has variance of $p(1-p)$ that is maximized at 0.25 when $p = 0.5$, i.e. when the model is maximally uncertain about preference label. Under the zero-mean Gaussian noise assumption, we change the uncertainty representation from Bernoulli variance to Gaussian variance, which we specify using constant covariance matrix of $\mathbf{V} = 0.07 \cdot \mathbf{I}$. This roughly perturbs the predicted BT likelihood of preferring $\tau_a \succ \tau_b$ with probability of 0.07 to account for label error, thus preventing the model from making large updates towards overly confident predictions.

### A.1.3 On EKF hyperparameters:

The main hyperparameters of Kalman filters are the dynamics noise covariance $\mathbf{U} \in \mathbb{R}^{|\boldsymbol{\theta}|\times|\boldsymbol{\theta}|}$, the measurement noise covariance $\mathbf{V} \in \mathbb{R}^{|y|\times|y|}$, and the belief initialization covariance $\mathbf{W} \in \mathbb{R}^{|\boldsymbol{\theta}|\times|\boldsymbol{\theta}|}$ in the initial belief $\boldsymbol{b}^0 = p(\boldsymbol{\theta}_0) = \mathcal{N}(0, \mathbf{W})$. As the goal of applying Bayesian filters to train neural networks is to enable sequential learning from potentially non-stationary data without overfitting to data it has seen so far, we apply weak parameter regularization by specifying small dynamics noise of $\mathbf{U} = 0.0001 \cdot \mathbf{I}$; this serves to continuously apply weak perturbation to model parameters so as to prevent overfitting. On the other hand, to enable model learning via posterior updates, we set prior noise to moderate level of $\mathbf{W} = 0.07 \cdot \mathbf{I}$.

We apply measurement noise to deal with potentially noisy data, which in the domain of learning from pairwise preferences amounts to dealing with flipped preference labels, e.g., among two trajectories, $\tau_a$ is supposed to be the better trajectory, but an annotator mistakenly indicated $\tau_b$ as the preferred item. In our synthetic label experiments, we set about $5\% - 10\%$ of our generated labels as flipped for each task. We set our measurement noise covariance $\mathbf{V} = 0.07 \cdot \mathbf{I}$ which roughly perturbs the predicted BT likelihood of preferring $\tau_a \succ \tau_b$ with probability of 0.07 to account for label error. Since the BT model is inherently a Bernoulli distribution where a correct preference label prediction only requires predicted probability of over 50%, we believe our chosen measurement noise is of appropriate scale.

All noise hyperparameters were swept roughly on a $\log_3$ scale. We found PreferenceEKF's performance to be sensitive to all noise hyperparameters. For example, large $\mathbf{W}$ and small $\mathbf{V}$ would lead to very strong posterior updates, leading to overfitting behavior where test-likelihood would increase for a few queries before steady decline. On the other extreme, small $\mathbf{W}$ and large $\mathbf{V}$ would lead to weak posterior updates, causing underfitting behavior where test-likelihood barely sees any improvement. For $\mathbf{U}$, we found that values much higher than $\mathbf{U} = 0.0001 \cdot \mathbf{I}$ prevented model from learning altogether, while tiny values lead to numerical instability.

## A.2 Preference-based Reward Learning

### A.2.1 Statistical testing

To provide statistical significance to the main claims from Section 5.1, we conduct hypothesis testing of 1) whether the active variant of each algorithm outperforms its random variant and 2) whether active PreferenceEKF outperforms active variants of other Bayesian deep learning baselines. For the summary statistic of each active reward learning experiment run, we compute the normalized area under curve (AUC) of the log-likelihood plot in Figure 1a. This measures the rate of improvement for log-likelihood.

Since all runs from Figure 1a are performed using the same set of 12 random seeds and the same train/test dataset split, we conduct our hypothesis testing using one-sided bootstrap hypothesis test to compare the normalized AUC between two sets of runs. We additionally compute the 95% confidence interval as well as Cohen's $d$ for effect size. In the first 5 rows of Table 1, we show the performance of active versus random variant of each algorithm. We see that active DeepEnsemble, Laplace and LLMCMC outperform their random

counterparts in normalized AUC with high statistical significance, and Dropout completely fails to do so. While PreferenceEKF does outperforms its random counterparts on average according to Figure 1a, it does so with low statistical significance. We thus conclude that active PreferenceEKF performs on par with its random variant, but is unable to outperform it.

In the last 4 rows of Table 1, we show the performance of active PreferenceEKF versus active variant of other baselines. We see that active PreferenceEKF outperforms active variants of all baselines in normalized AUC with high statistical significance, with exception of LLMCMC, where their performance were on par with each other. Compared to LLMCMC, our method requires much less time to perform posterior inference (see Figure 1b) and does not require storage of all queries seen so far, which we see as major advantages despite similar downstream preference learning performance.

| Test | mean diff | $p$-value | Cohen's $d$ | 95% CI |
|------|-----------|-----------|-------------|--------|
| EKF (A vs. R) | 0.01 | 0.077 | 0.59 (medium) | $(0.00, \infty)$ |
| DeepEnsemble (A vs. R) | 0.12 | $< 0.001$ | 4.93 (large) | $(0.10, \infty)$ |
| Dropout (A vs. R) | -0.02 | 0.825 | -0.36 (small) | $(-0.05, \infty)$ |
| Laplace (A vs. R) | 0.04 | 0.011 | 0.90 (large) | $(0.01, \infty)$ |
| LLMCMC (A vs. R) | 0.03 | $< 0.001$ | 1.92 (large) | $(0.02, \infty)$ |
| EKF vs. DeepEnsemble | 0.05 | $< 0.001$ | 2.26 (large) | $(0.04, \infty)$ |
| EKF vs. Dropout | 0.25 | $< 0.001$ | 5.23 (large) | $(0.21, \infty)$ |
| EKF vs. Laplace | 0.21 | $< 0.001$ | 5.17 (large) | $(0.18, \infty)$ |
| EKF vs. LLMCMC | 0.01 | 0.064 | 0.57 (medium) | $(0.00, \infty)$ |

Table 1: One-sided bootstrap tests comparing active vs. random variants of each algorithm, and active EKF vs. active variant of other baseline algorithms.

### A.2.2 Runtime experiments

Due to readability of runtime scale, we provide the table version of Figure 1b here:

Table 2: Runtime in minutes. Table version of Figure 1b.

| Runtime | PreferenceEKF | DeepEnsemble | Dropout | Laplace | LLMCMC |
|---------|---------------|--------------|---------|---------|--------|
| Active | $12.1 \pm 0.1$ | $97.7 \pm 0.1$ | $62.1 \pm 0.0$ | $354.6 \pm 2.0$ | $780.2 \pm 40.2$ |
| Random | $9.9 \pm 0.1$ | $97.5 \pm 0.7$ | $60.8 \pm 0.2$ | $354.1 \pm 0.7$ | $571.5 \pm 11.6$ |

Similarly, we provide the table version of Figure 2b here:

Table 3: Final likelihood vs. ensemble size $M$. Table version of Figure 2b.

| Final Likelihood | PreferenceEKF | DeepEnsemble | Dropout |
|------------------|---------------|--------------|---------|
| $M=5$ | $-0.220 \pm 0.038$ | $-0.518 \pm 0.055$ | $-0.774 \pm 0.022$ |
| $M=15$ | $-0.292 \pm 0.022$ | $-0.538 \pm 0.114$ | $-0.612 \pm 0.138$ |
| $M=30$ | $-0.267 \pm 0.020$ | $-0.462 \pm 0.095$ | $-0.704 \pm 0.130$ |
| $M=50$ | $-0.249 \pm 0.050$ | $-0.508 \pm 0.082$ | $-0.823 \pm 0.035$ |
| $M=75$ | $-0.279 \pm 0.024$ | $-0.477 \pm 0.100$ | $-0.680 \pm 0.220$ |
| $M=100$ | $-0.255 \pm 0.018$ | $-0.308 \pm 0.047$ | $-0.721 \pm 0.168$ |
| $M=150$ | $-0.223 \pm 0.000$ | $-0.368 \pm 0.068$ | $-0.629 \pm 0.037$ |

### A.2.3 Implementation details

Unless otherwise stated, all reward learning experiments are done using subspace dimensionality $|\boldsymbol{z}| = 200$, query budget $B = 60$, and partial trajectory of length 50. All neural networks reward model are represented using multi-layer perceptrons (MLP) with two hidden layers of 64 units. We apply normalization to all input features. PreferenceEKF and Dropout uses $M = 100$ model parameter samples to compute the acquisition function and posterior predictive distribution, while DeepEnsemble trains $M = 5$ independent networks, each with different weight initialization and randomness for minibatch shuffling.

All tasks use a pool of 150K pairwise partial trajectory queries drawn from the trajectory dataset to perform random or active querying over, and 3000 test queries for log-likelihood evaluation. For generation of noisy-optimal synthetic labels, we apply trajectory return normalization before passing trajectory pairs through the BT model (Eq. 1) to compute the likelihood $p_\theta(\tau_a \succ \tau_b)$. We use temperature parameter of $\beta = 7$, resulting in roughly 5-15% mistaken preference labels per task.

Before the sequential learning phase starting on Line 13, all algorithms receive a small dataset consisting of $\tau = 8$ query-response pairs for belief initialization, i.e., all algorithms observe a total of $\tau + B = 8 + 60 = 68$ samples. All algorithms run variants of gradient descent (GD) on the warm-up dataset for 420 optimizer steps. While PreferenceEKF uses SGD with learning rate of 1e-4, momentum of 0.9, and batch size of 1, DeepEnsemble and Dropout uses Adam (Kingma & Ba, 2014) with learning rate of 1e-4 along with default hyperparamters from Optax (DeepMind et al., 2020), and batch size of 8.

PreferenceEKF constructs the subspace by running SVD on the GD iterates obtained from running SGD on the warmup dataset. We throw away the first 20 out of the 420 GD iterates and keep only every other remaining iterate, for a total of $(420 - 20)/2 = 200$ iterates. Thus, SVD takes in a model parameter array of shape $(200 \times |\boldsymbol{\theta}|)$, and return a projection matrix $\mathbf{A}$ of shape $(200 \times |\boldsymbol{z}|)$ by keeping only the top $|\boldsymbol{z}| = 200$ principal components. The final GD iterate is used as the full space parameter offset $\boldsymbol{\theta}_*$, which, along with projection matrix $\mathbf{A}$, is used to transform from the subspace back up to the full space for, e.g. computing predictive distributions as described in Section 4. Finally, PreferenceEKF performs belief initialization (12) in the subspace using a zero-mean isotropic Gaussian of dimension $|\boldsymbol{z}| = 200$.

On the belief update step (Line 16), PreferenceEKF learns from only the most recent query-label pair, while DeepEnsemble and Dropout learns learns from all data seen so far. Note that the specific filtering algorithm we use is the iterated EKF (Bell & Cathey, 1993), which repeatedly re-linearize the measurement model around the estimated posterior. Empirically, we observed better log-likelihood evaluation performance in exchange for marginally extra runtime. We refer to the number of such re-linearization steps on every new sample as $n_{\text{linearize}}$. For further details on iterated EKF, refer to Section 8.3.2.2 of Murphy (2023b). We use $n_{\text{linearize}} = 5$ for our experiments, but found that the performance of PreferenceEKF to be relatively robust for this hyperparameter. We detail our choice of prior, dynamics, and observation noise levels in Section A.1.3.

**On methods for subspace construction** The SVD-based approach and the random projection approach are the two primary methods for neural network subspace construction studied across literature (Izmailov et al., 2020; Larsen et al., 2022). Our default implementation of PreferenceEKF uses the SVD-based subspace construction method, where we first run SGD on an initial preference labeled dataset, then apply SVD on the SGD iterates to obtain a subspace projection matrix. We also experimented with using the Adam optimizer instead of SGD to produce the iterates, but found this to lead to poor empirical performance. This is consistent with previous works which found that SGD with a high constant learning is crucial to producing parameter iterates with enough variance to construct a subspace effective for optimization and inference (Fort et al., 2020). We hypothesize that Adam's per-parameter learning rate adaptation scheme results in more performant loss minimization but less varied parameter iterates across the optimization trajectory, thus producing a subspace that does not span the full parameter space enough for effective inference.

As an alternative to the SVD-based subspace construction approach, the projection matrix can be obtained via random projections by computing $\mathbf{A} \in \mathbb{R}^{|\boldsymbol{\theta}| \times |\boldsymbol{z}|}$ as a random Gaussian matrix with columns normalized to 1 (Li et al., 2018). See Section 5.4 for a study comparing the two approaches. See also Section A.2.8 for a

usage of the random projection method for cases where we we don't have access to an initial dataset, thus removing PreferenceEKF's usage of SGD-based initialization altogether.

We additionally note that PreferenceEKF's early performance upon belief initialization, prior to the active learning / random sampling phase, is often much higher compared to all baseline methods. We hypothesize that this is due to PreferenceEKF using SGD only as a mean to construct the subspace projection matrix, but the actual belief is initialized as a zero-mean Gaussian in this learned subspace. Compared to methods that rely heavily on SGD such as DeepEnsemble, Dropout, and Laplace, the subspace approach may simply be less overfitted to the initial dataset.

### A.2.4 Baseline algorithms

The primary tradeoff that Bayesian deep learning (BDL) algorithms are concerned with is the computational tractability and approximation quality of the posterior distribution over model parameters given data $p(\boldsymbol{\theta} \mid \mathcal{D})$. We selected DeepEnsemble and Dropout as baselines due to 1) their popularity for representing uncertainty in neural networks and 2) their simplicity in that they only rely on standard neural network training techniques such as SGD and dropout, without any classic Bayesian inference algorithms. We selected Laplace and LLMCMC as they represent state of the art works in scaling classic inference algorithms to the high-dimensional parameter space of neural network training.

For high-dimensional models such as neural networks, the posterior can be highly multi-modal, which can be difficult to approximate for algorithms that use unimodal distributions (typically Gaussian) such as Laplace approximation and extended Kalman filters. On the other hand, while Markov chain Monte Carlo (MCMC) has been the gold standard for posterior approximation (Izmailov et al., 2021), they are very difficult to scale to large models with many parameters. As such, many BDL algorithms try to "be Bayesian" over only a subset or subspace of model parameters, or rely on ensembling to hopefully reach multiple posterior modes. Here we provide a high-level description of the five classes of BDL algorithms we use for our experiments, how they perform belief initialization (Line 12) and belief update (Line 16), the corresponding implementation details, as well as where they have been used in the reward learning literature.

**DeepEnsemble** and **Dropout** are among the most widely-used BDL algorithms for reward modeling and more generally, uncertainty quantification in neural networks (Christiano et al., 2017; Gleave & Irving, 2022; Chen et al., 2020; Hoque et al., 2022; Jaques et al., 2019). They approximate the posterior by relying on randomness (e.g., weight initialization, mini-batch sampling order) to train multiple models and average over their predictions. While DeepEnsemble has the computational burden of actually training multiple neural networks, Dropout masks out subset of model parameters during training and computes the posterior predictive distribution by averaging predictions from multiple model copies with different weight masks during inference time, thus requiring training of only one model. The idea for both approach is for the multiple resulting models to act as samples from the posterior distribution. All $M$ models trained under DeepEnsemble method receive a different stream of mini-batches for training. Dropout uses weight dropout probability of 0.3 for all experiments, during both training and inference. For both methods, belief initialization is done by running SGD on an initial dataset, and belief update is performed by running SGD on all data seen so far.

**Laplace**: While Laplace approximation (LA) has traditionally been used for smaller models in logistic regression and Gaussian process-based regression models (Biyik et al., 2020; Rasmussen & Williams, 2005), recent advancements such as those in Dangel et al. (2025); Daxberger et al. (2024) have made the technique highly scalable to neural network architectures. Combined with parameter efficient finetuning technique such as LoRa (Hu et al., 2021), LA has even been applied to transformer-scaled reward models (Yang et al., 2024a). By approximating likelihood curvature around a model solution trained via maximum likelihood methods such as gradient descent, LA constructs a local Gaussian approximation to the model posterior. We use the full curvature approximation-based approach of Weber et al. (2025) to perform LA over the entire reward model, with prior precision value of 1000. Once the curvature information has been constructed for the Gaussian posterior approximation, we can sample arbitrary number of model parameters from the posterior. Both belief initialization and belief update are done by first running SGD on all data seen so far, then performing LA on the final SGD iterate.

**LLMCMC**: Despite the high quality posterior approximation of MCMC methods for smaller models such as linear models (Bıyık et al., 2020; Hadfield-Menell et al., 2017), they have are not widely used for neural network posterior inference due to their poor scalability to parameter count. Most application of MCMC to BDL trains the entire NN model using more efficient maximum likelihood methods like gradient descent, then perform MCMC only over the parameters of the final layer. We chose this "last-layer Bayesian" approach as it has been shown to strike a good balance between computational tractability and approximation quality (Brown et al., 2020; Snoek et al., 2015). The specific MCMC sampler we use is NUTS (Hoffman & Gelman, 2014). On active learning step, we construct a new log-density function using the aggregated dataset using all samples seen so far. For belief initialization, we use 500 warm up MCMC iterations followed by 500 additional iterations. For belief update steps, since the log-density function should not differ too much with one additional aggregated sample, we set warm up iteration to be 20, followed by 500 additional iterations. We then subsample $M$ models from the resulting MCMC iterates to form our sampling-based posterior.

**PreferenceEKF**: While the preceding described methods perform optimization and inference over either the full model parameter set or a subset thereof, PreferenceEKF find a low dimensional subspace (as opposed to just a subset of the parameters) within the full parameter space, and perform inference within the subspace. The main insight of subspace inference approaches (Daxberger et al., 2021) is that due to the overparameterized nature of neural networks, capturing posterior information only over a constrained subspace would be a sufficient alternative to posterior inference over the whole network. Once a Gaussian approximation is obtained via subspace Kalman filtering, we can sample arbitrary number of model parameters from the posterior.

### A.2.5 Acquisition functions

The InfoGain acquisition function introduced in Eq. 2a was developed by Bıyık et al. (2020) for active reward learning using linear reward models. To motivate its origin, we first express the InfoGain objective in three equivalent forms below due to symmetry of mutual information.

$$Q_i^* = \arg\max_{Q_i} I\left(\boldsymbol{\theta}; y_i \mid Q_i, \boldsymbol{b}^{i-1}\right) \tag{9a}$$

$$= \arg\max_{Q_i} H\left(\boldsymbol{\theta} \mid Q_i, \boldsymbol{b}^{i-1}\right) - \mathbb{E}_{y_i}\left[H(\boldsymbol{\theta} \mid y_i, Q_i, \boldsymbol{b}^{i-1})\right] \tag{9b}$$

$$= \arg\max_{Q_i} H\left(y_i \mid Q_i, \boldsymbol{b}^{i-1}\right) - \mathbb{E}_{\boldsymbol{\theta}}\left[H(y_i \mid \boldsymbol{\theta}, Q_i)\right], \tag{9c}$$

where $\boldsymbol{b}^{i-1} = p(\boldsymbol{\theta} \mid \mathcal{D}_{1:i-1})$ is the posterior distribution over RM parameters after learning from $(i-1)$ queries. The idea of mutual information-based acquisition functions is rooted in the concept of expected information gain studied in Bayesian optimal experiment design and active data selection (MacKay, 1992; Lindley, 1956). It was later extended to Bayesian optimization using Gaussian process models under the methods Bayesian active learning by disagreement (BALD) (Houlsby et al., 2011), entropy search (ES) (Hennig & Schuler, 2012), and predictive entropy search (PES) (Hernández-Lobato et al., 2014). In particular, the mutual information objective function in Eq. 9a is expressed in its ES form in Eq. 9b, and expressed in its equivalent but computationally efficient PES form in Eq. 9c.

Our PreferenceEKF method focuses on efficient sampling of high-dimensional neural network model parameters to approximate the predictive distribution for optimizing Eq. 9c, which we derive as follows. We refer to Section 5 of Bıyık et al. (2020) for further interpretations of the objective, and Section 9.1 of their work for derivation of the sampling-based approximation shown in Eq. 3.

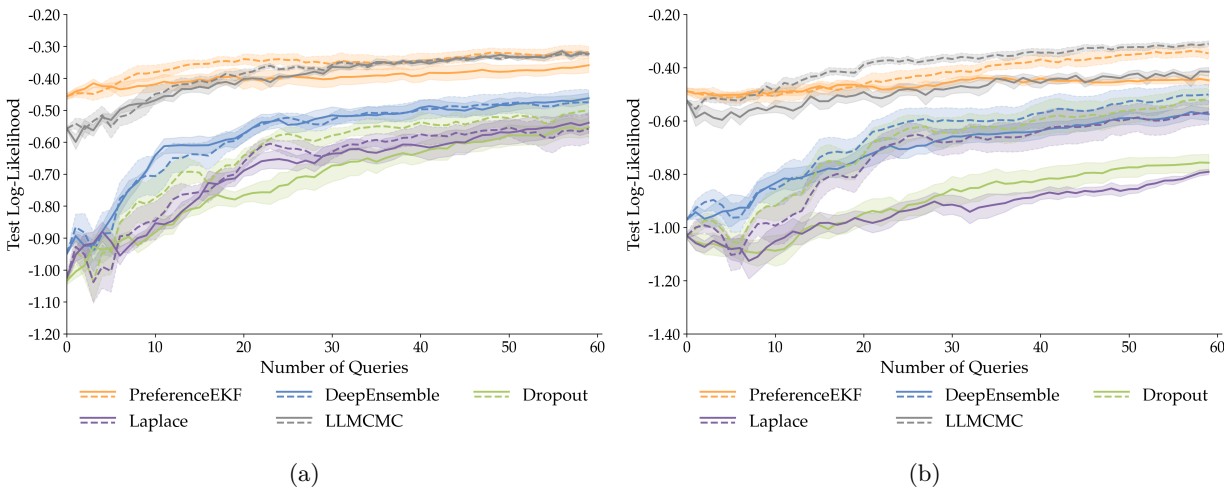

Figure A.1: Figure A.1a shows log-likelihood comparison of the random (dashed line) and active (solid line) variants of each algorithm using the disagreement acquisition function (higher means better fitting of annotator preference distribution). Figure A.1b shows the results for entropy-based acquisition function.

$$Q_i^* = \arg\max_{Q_i} I\left(\boldsymbol{\theta}; y_i \mid Q_i, \boldsymbol{b}^{i-1}\right)$$

$$= \arg\max_{Q_i} H\left(y_i \mid Q_i, \boldsymbol{b}^{i-1}\right) - H\left(y_i \mid \boldsymbol{\theta}, Q_i, \boldsymbol{b}^{i-1}\right)$$

$$= \arg\max_{Q_i} H\left(y_i \mid Q_i, \boldsymbol{b}^{i-1}\right) - \mathbb{E}_{\boldsymbol{\theta} \sim p(\boldsymbol{\theta}\mid\boldsymbol{b}^{i-1})}\left[H\left(y_i \mid \boldsymbol{\theta}, Q_i\right)\right]$$

$$= \arg\max_{Q_i} H\left(y_i \mid Q_i, \boldsymbol{b}^{i-1}\right) - \mathbb{E}_{\boldsymbol{\theta}}\left[H(y_i \mid \boldsymbol{\theta}, Q_i)\right]$$

Although our main experiments all use the InfoGain acquisition function to showcase the advantage of being able to sample from high-dimensional neural network parameter distributions, the PreferenceEKF method is agnostic to the acquisition function used for active learning. While Figure 1a and Figure A.2 showcase the aggregate and per-task log-likelihood results for active preference-based reward learning experiments using InfoGain, here we show additional results using two more commonly-used acquisition functions, disagreement and entropy. Disagreement selects the query $Q_i$ for which the predicted preference label $\mathbb{1}(\tau_a^i \succ \tau_b^i)$ has the highest variance across the ensemble or sampled models, and entropy selects the queries for which the Bradley-Terry posterior predictive distribution exhibits the highest entropy.

We show in Figure A.1 that both disagreement and entropy acquisition function resulted in similar trends, where although PreferenceEKF and LLMCMC perform the best overall, neither algorithm's active learning variant outperformed their random counterpart. This is contrast to the InfoGain acquisition function result in Figure 1a, where all algorithm's active variant outperformed their random variant. This demonstrates that while PreferenceEKF and LLMCMC prove to be the most effective at learning a posterior for fitting the annotator's preference distribution, the choice of acquisition function still matters greatly for sample-efficient active learning, with InfoGain being the best performing acquisition function overall, followed by disagreement and then entropy. We further show per-task preference-learning results for InfoGain, disagreement, and entropy in Figure A.2, Figure A.3, and Figure A.4, respectively.

### A.2.6 Reward learning using same number of model samples

In our main experiment results, while DeepEnsemble trains $M = 5$ separate reward models and use them to approximate the posterior, all other algorithms explicitly learn a posterior distribution over model parameters,

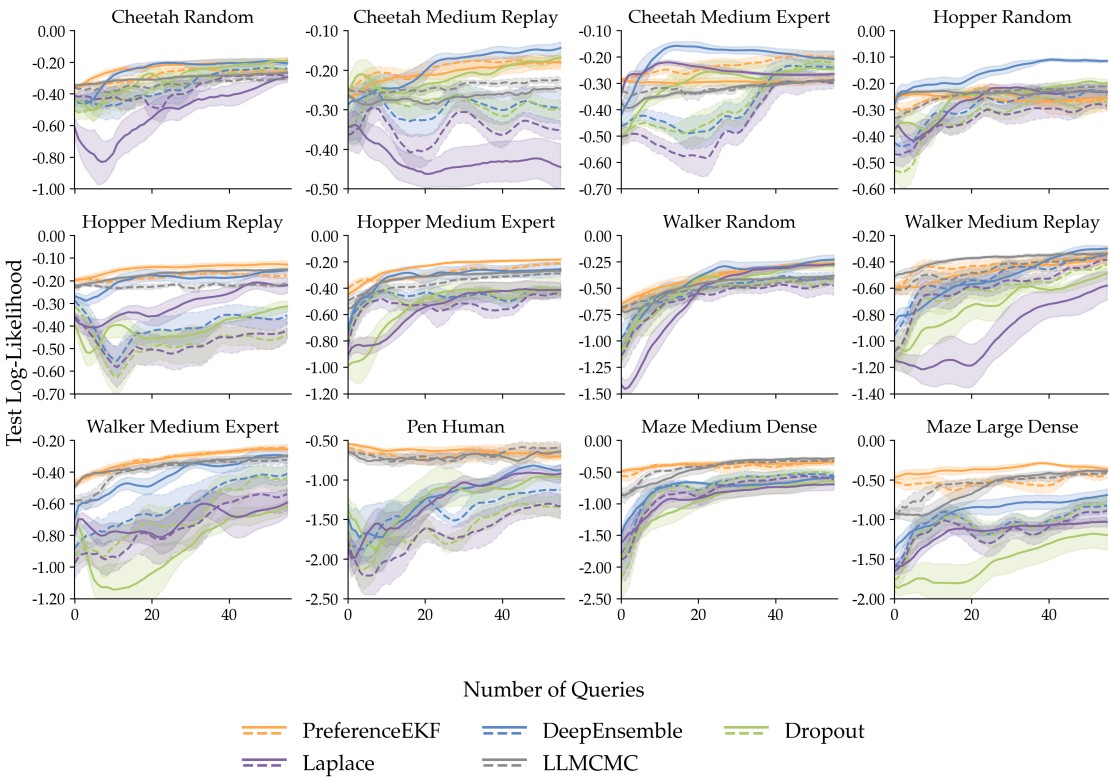

Figure A.2: Per-task reward learning performance using InfoGain acquisition function: comparison of the random (dashed line) and active (solid line) variants of the algorithms across 12 D4RL tasks (mean±s.e. over 5 seeds). In all tasks, active PreferenceEKF either performs on par with or outperforms other algorithms in terms of sample-efficiency and final log-likelihood.

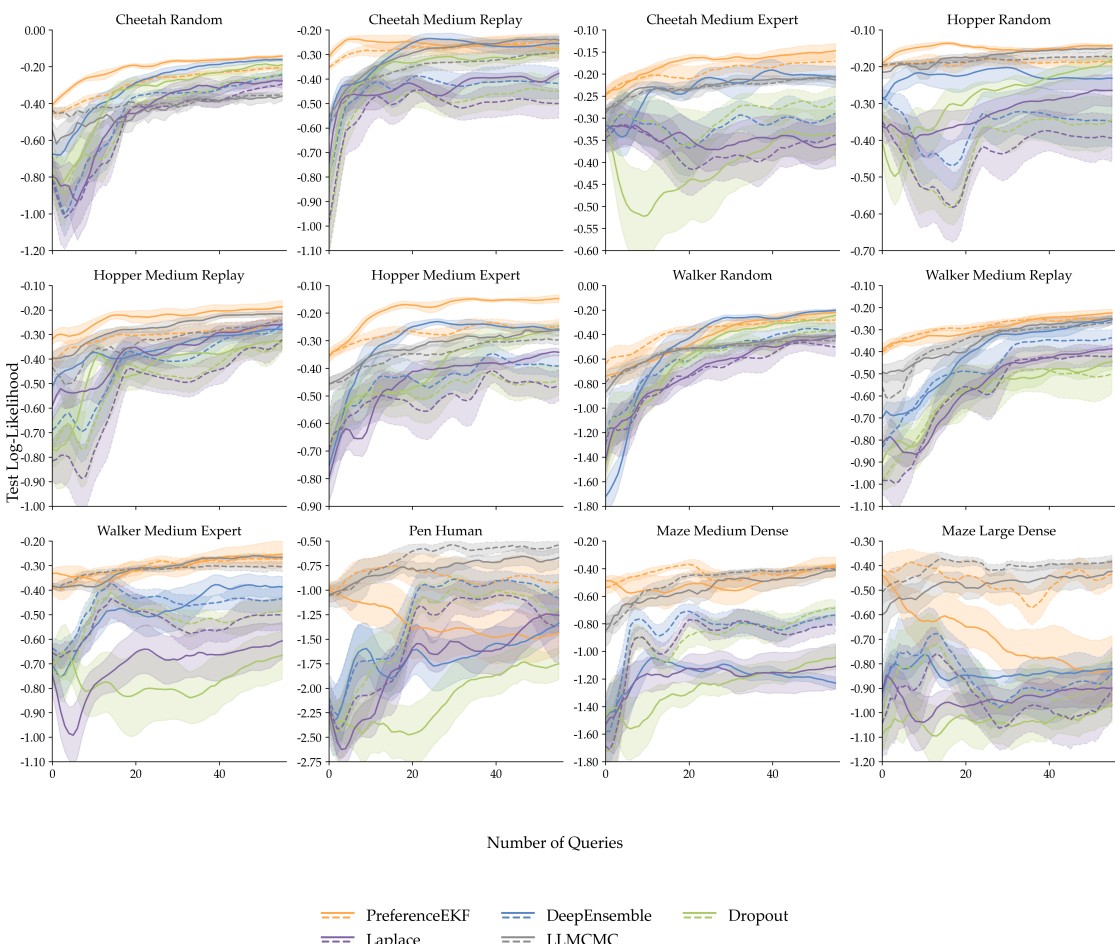

Figure A.3: Per-task reward learning performance using the disagreement acquisition function: comparison of the random (dashed line) and active (solid line) variants of the algorithms across 12 D4RL tasks for preference-based reward modeling (mean±s.e. over 5 seeds). In most tasks, active PreferenceEKF either performs on par with or outperforms other algorithms in terms of sample-efficiency and final log-likelihood. Pen Human are Maze Large Dense are particular outlier cases where active PreferenceEKF severely underperforms, which explains why the aggregate results in Figure A.1a look unfavorably for active PreferenceEKF relative to its random variant.

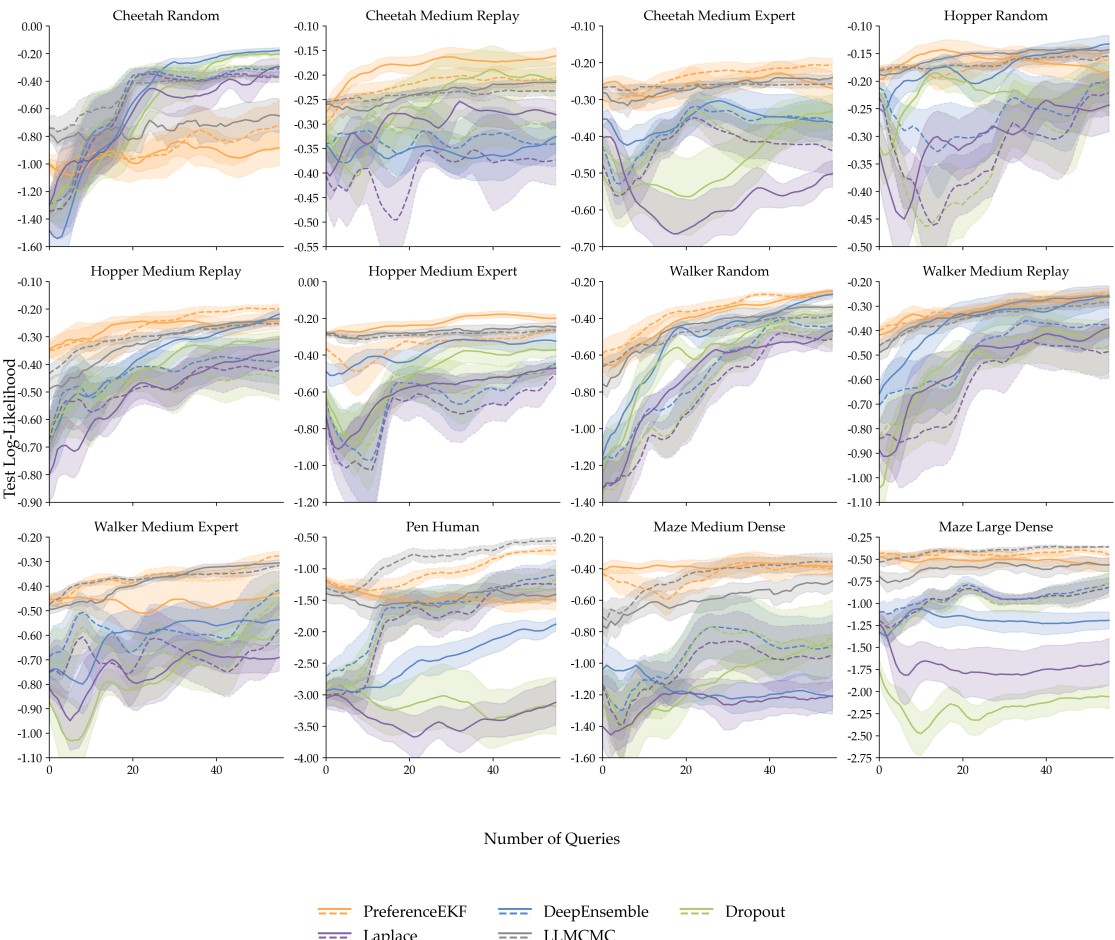

Figure A.4: Per-task reward learning performance using the entropy acquisition function: comparison of the random (dashed line) and active (solid line) variants of the algorithms across 12 D4RL tasks for preference-based reward modeling (mean±s.e. over 5 seeds).

and can thus sample an arbitrary number of model parameters for computing acquisition functions and making predictions; our experiments used $M = 100$.

This raises the question of whether PreferenceEKF's superior preference learning sample efficiency is solely due to the larger number of posterior samples, or whether the learned posterior indeed captures the annotator's preference. To investigate this, we set $M = 5$ for all algorithms and see in Figure A.5 that PreferenceEKF still outperforms all methods in test log-likelihood. This signals that higher model sample count is not the only factor that can explain PreferenceEKF's superior sample efficiency, and that the subspace approach for uncertainty representation indeed results in a learned posterior that captures the annotator's preferences well.

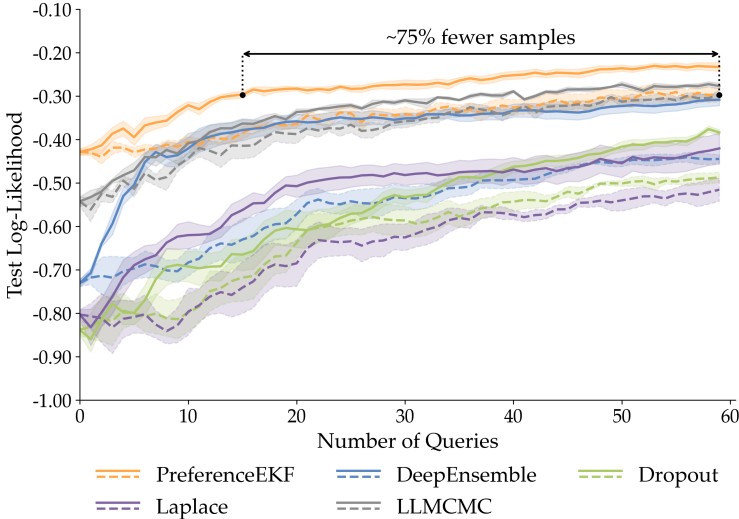

Figure A.5: Task-aggregate reward learning performance with all algorithms using the same number of model samples ($M = 5$). Runs are averaged over all 12 tasks of 5 seeds each.

### A.2.7 Training runtime under the same compute budget

Section 5.2 showed that PreferenceEKF had much faster training run time compared to the other methods. The setting was biased to be favorable towards PreferenceEKF, as it only needs to perform one belief update step on the most recent query (thanks to its sequential learning nature), whereas all the other methods needs to perform multiple gradient descent updates on all data seen so far so as to ensure that SGD converges to a maximum likelihood solution.

To ensure a more fair runtime comparison. We set the number of SGD iterations for belief update of all baselines to match the compute budget of EKF's belief update step, and additionally set the number of model samples for all methods to be $M = 5$ to match that of DeepEnsemble. We show in Figure A.6a that under the setting with reduced number of SGD steps, the methods that rely on SGD (DeepEnsemble, Dropout, Laplace) failed to find converge, and were thus unable to fit the preference distribution as indicated by low log-likelihood. PreferenceEKF and LLMCMC were still able to fit the preference distribution, with the former still retaining a clear lead in having the fastest training runtime as shown in Figure A.6b.

### A.2.8 Reward learning without an initial dataset

In Section 4, it was shown that subspace construction for PreferenceEKF can be done via either SVD on SGD iterates trained on an initial dataset, or random projections. Algorithm 1 further indicates that regardless of the subspace construction method, PreferenceEKF still relies on access to an initial dataset of already labeled queries.

It is desirable for any reward learning algorithms to still work in domains where such initial dataset is unavailable. To investigate the reliance of all methods on access to initial data, we show in Figure A.7 reward

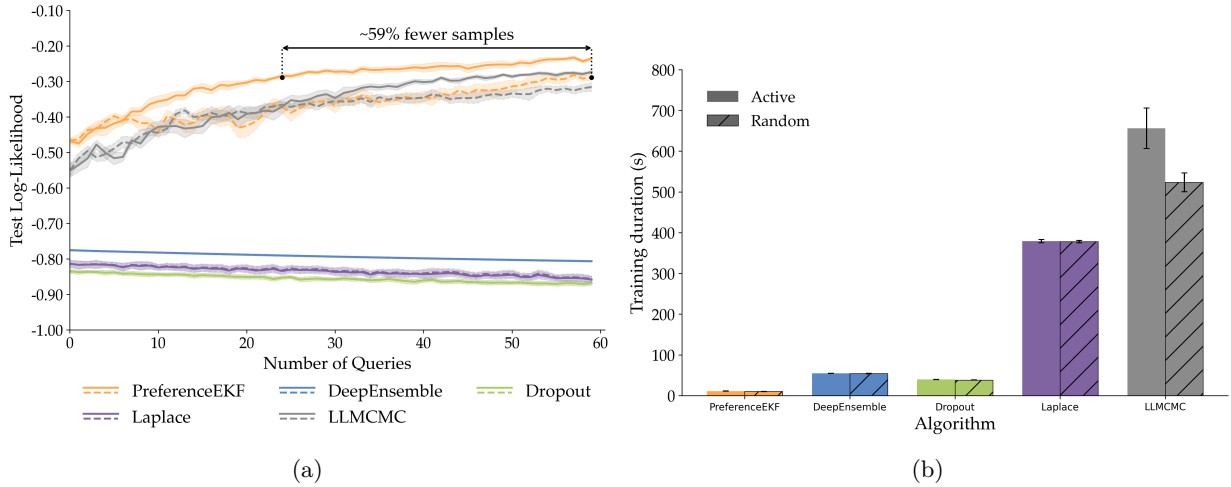

(a)  (b)

Figure A.6: Figure A.6a shows log-likelihood comparison of the random (dashed line) and active (solid line) variants of each algorithm using the InfoGain acquisition function, with reduced belief update compute budget. We see that all methods that rely heavily on SGD fail to learn, while PreferenceEKF and LLMCMC retain good performance. Figure A.6b shows training runtime duration of both active and random variant of each algorithm. Each line plot and bar plot is aggregated over 12 D4RL tasks (mean±s.e. over 5 seeds).

learning results where we remove access to any initial data, thus making posterior updates possible only with an annotator in the loop. In this case, PreferenceEKF uses the random projection method for subspace construction. We see that PreferenceEKF still outperforms all baselines, showcasing its flexibility in learning reward models even without access to existing labeled queries.

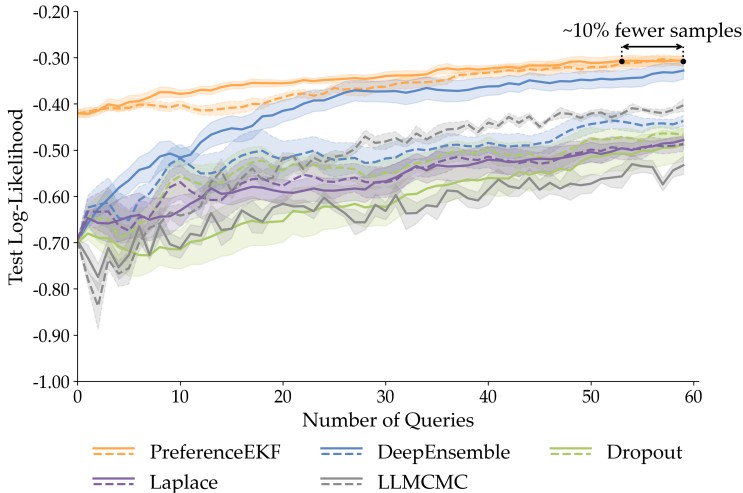

Figure A.7: Task-aggregate reward learning performance with all algorithms using no initial dataset, thus no warmup SGD phase. Runs are averaged over all 12 tasks of 5 seeds each (mean±s.e.).

### A.2.9  Reward learning from human annotators

Our main experiments are conducted exclusively with synthetic oracle preference labels, where of the two trajectories being compared, the trajectory with higher summed reward is designated as the preferred trajectory. To test out the methods' ability to learn rewards from real human preferences, we use the crowd-sourced preference labels from Yuan et al. (2023) to perform reward learning. We show in Figure A.8

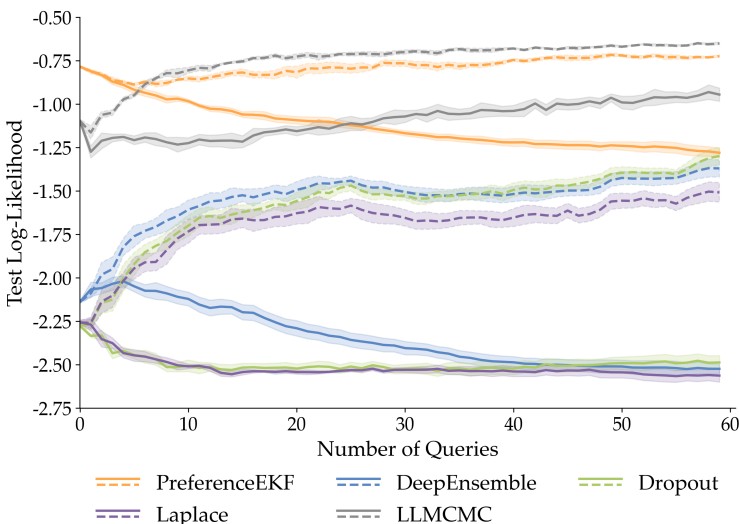

Figure A.8: Task-aggregate reward learning performance with all algorithms using crowd-sourced real human preference labels. Runs are averaged over all 12 tasks of 5 seeds each (mean±s.e.).

that no methods reached great log-likelihood evaluation, and none of the methods' active variant was able to outperform their random variant. This is likely due to the crowd-sourced nature of the labels, which may induce multi-modal preference distribution underlying the labels that may be difficult for our benchmark algorithms to capture. We emphasize that our work's main contribution is a sample-efficient active reward learning algorithm for the single annotator setting, and we leave adaptation of our work to multi-annotator settings to future work.

### A.2.10  Reward learning from real robotics data

We extend the application of our method to real-world robotics dataset, where we leverage sparse binary task success or failure signal as preference feedback. This setting is common in recent robot reward model works such as Yang et al. (2024b) and Liang et al. (2026). We use the rollout datasets from SOAR (Zhou et al., 2024), where the trajectories are collected by a fleet of 5 WidowX robot arms over a variety of manipulation tasks, such as putting a blue block in a wooden bowl or transferring a mushroom from a bowl to a table. The trajectory observation data are 7-dimensional proprioceptive states encoding end effector translation (XYZ) and rotation (roll, pitch, yaw), along with gripper open/close state (scalar). For a given task, we generate preference labels by sampling one successful trajectory and one failed trajectory, and label the successful trajectory as the preferred one.

Since the SOAR dataset was primarily collected for the purpose of learning from suboptimal data free of human supervision in the real world, and our pairwise sampling procedure requires (success, failed) trajectory pairings, we found that many task datasets from SOAR contained either 1) too few trajectories in total or 2) way more failed trajectories than successful ones. We narrowed down our evaluation task suite down to 3 tasks that contained sufficient number of (success, failed) pairings, and evaluated all methods on all tasks. All trajectories are of fixed length of 100 steps, with no partial segment sampling as we do in our main results in Section 5.1. In Figure A.9, we show that PreferenceEKF and LLMCMC both achieve the best performance over all methods considered, showcasing the applicability of our method to real world robotics data. We further show per-task performance in Figure A.10.

### A.2.11  Reward learning from pixel data

While our main results in Section 5 are performed on state-based control tasks, here we showcase the applicability of PreferenceEKF to pixel-based tasks. We focus on the Visual D4RL (V-D4RL) benchmark (Lu

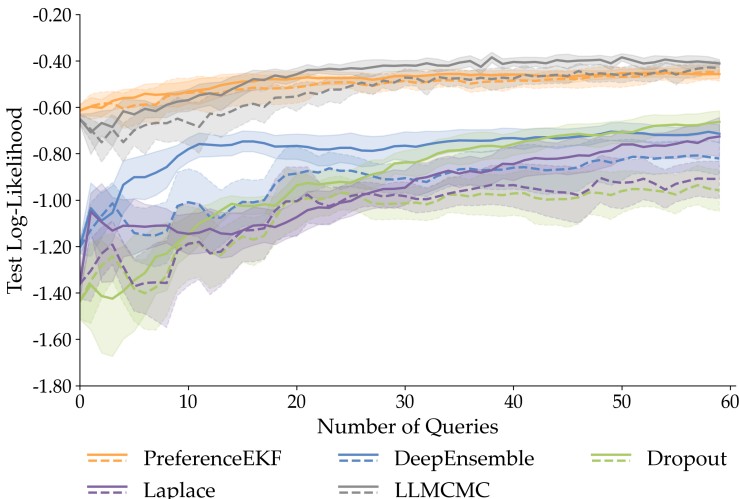

Figure A.9: Task-aggregate reward learning performance of all methods on the SOAR dataset, with random (dashed line) and active (solid line) variants of each method. Runs are aggregated over 3 tasks (mean ± 95% boostrap confidence interval over 12 seeds each task).

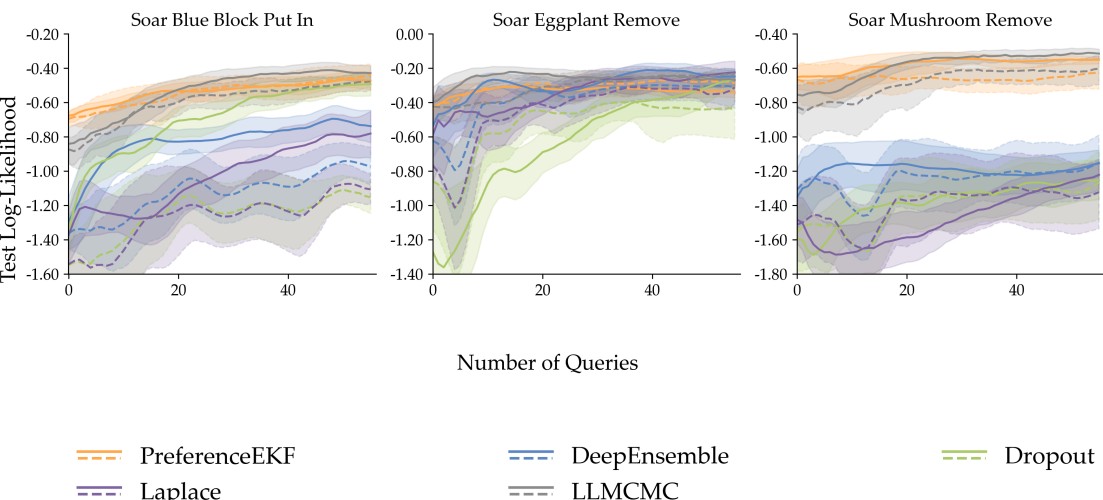

Figure A.10: Per-task reward learning performance of all methods on the SOAR dataset, with random (dashed line) and active (solid line) variants (mean ± 95% boostrap confidence interval over 12 seeds).

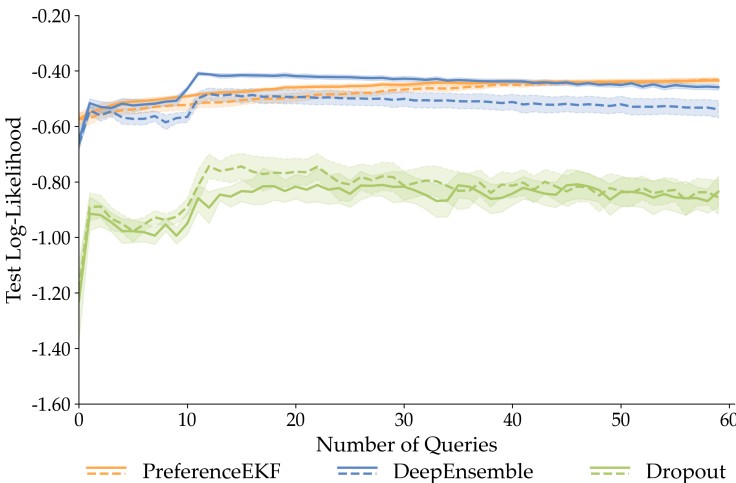

Figure A.11: Task-aggregate reward learning performance of PreferenceEKF on the pixel-based V-D4RL benchmark, with random (dashed line) and active (solid line) variants. Runs are aggregated over 3 pixel-based VD4RL tasks (mean±s.e. over 5 seeds).

et al., 2023), which contains rendered pixel-image observations corresponding to datasets from the state-based D4RL benchmark.

Our pixel-based reward model architecture consists of an ImageNet-pretrained ResNet18 image encoder with embedding dimension of 512 (Deng et al., 2009; He et al., 2016) as the backbone and a two-layer MLP with 256 hidden units per layer as the reward prediction head. We finetune the entire reward model via SGD as part of the belief initialization step of Line 12, and perform EKF inference within the subspace of only the reward head parameters while keeping the finetuned backbone frozen. We take a similar approach for the baseline methods, DeepEnsemble and Dropout; due to computational constraints, we did not include Laplace and LLMCMC. Due to the increased task and model complexity, we construct a subspace with dimensionality of 500 (compared to 200 in the state-based tasks with smaller reward models), and use random projection to do so since a larger subspace benefits equally from random projection versus SVD-based construction techniques as shown in Figure 3b.

Since EKF's belief update procedure scales cubically with dimensionality of the observation space, we use a measurement likelihood function (Eq. 4) over trajectory embeddings rather than raw trajectory pixels. We compute embeddings from the final layer of the ResNet18 backbone before the reward prediction head, and mean-pool the embeddings across all timesteps of a trajectory segment to obtain embeddings that aggregate trajectory-level information. Empirically, raw pixel observations over trajectory segment lengths of 10 steps with images of height, width, channel $(84, 84, 3)$ would result in observation dimension of $10 \times 84 \times 84 \times 3 = 211,680$ per trajectory, while mean-pooled embedding-based observation results in dimension of 512 per trajectory.

To finetune the pixel-based reward model which includes the entire ResNet18 backbone, we start with a much bigger initial query dataset of 150 (compared to just 8 in state-based experiments), and use a reduced learning rate of 0.0001 over 3000 mini-batches with batch size 16. In Figure A.11 and Figure A.12, we show that PreferenceEKF is indeed a viable method for active preference-based reward learning, and performs on par with DeepEnsemble while Dropout's performance suffers. While the performance of active versus random sampling varies across the three chosen pixel-based tasks, the active variant of PreferenceEKF as a whole shows promising improvement over the random variant. We leave research on EKF variants that efficiently scale with observation dimension, and more parameter efficient subspace inference methods such as those based on LoRa (Hu et al., 2021) to future work.

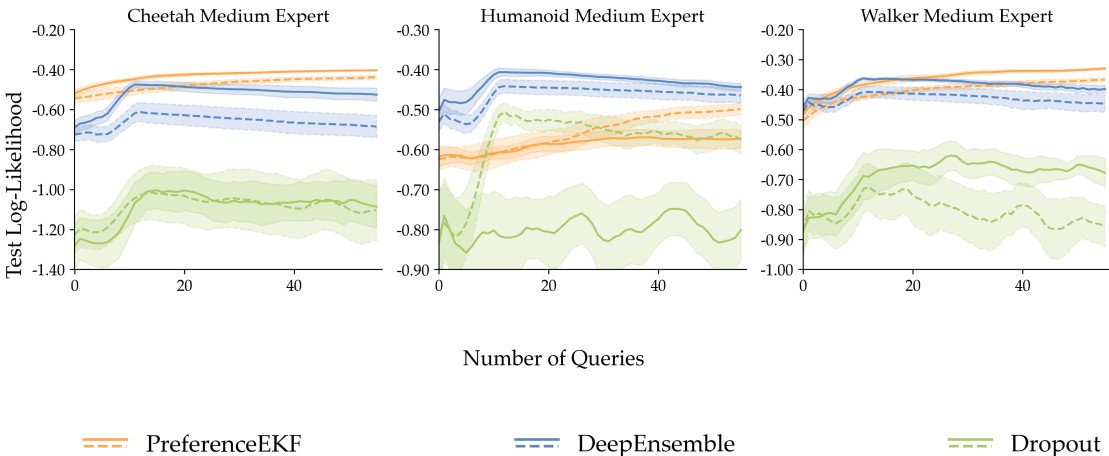

Figure A.12: Per-task reward learning performance of PreferenceEKF on the pixel-based V-D4RL benchmark, with random (dashed line) and active (solid line) variants. (mean±s.e. over 5 seeds).

### A.2.12    Model calibration experiments

In addition to the results from Section 5.3 on expected calibration error and Brier scores, we provide in Figure A.13 reliability diagrams computed from model predictions over all tasks and seeds. Due to the per-timestep parameterization of the reward model for computing the Bradley-Terry loss function Eq. 1, our binary preference query dataset is implemented to always have the second item be preferred over the first item. This corresponds to label of always 1, hence why the reliability diagrams only show calibration for half of the probability line. For both reliability diagram and expected calibration error, we discretize the $[0, 1]$ probability space into 10 bins. Upon inspection, we can see that PreferenceEKF and LLMCMC exhibit the lowest model calibration error.

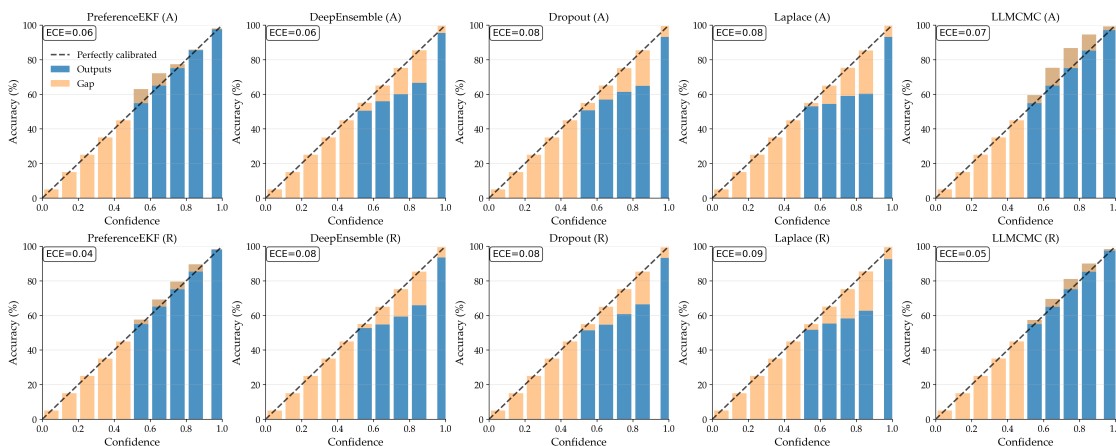

Figure A.13: Task-aggregate reliability diagram for all five methods' random and active variants. Diagrams are aggregated across all tasks adn seeds.

## A.3    Offline reinforcement learning

### A.3.1    Policy performance results

Figure A.14 shows comparison of policy optimization using the reward models learned from random and active variants of each algorithm, aggregated across 12 D4RL tasks in the offline RL setting (mean±s.e. over

5 seeds). Figure A.15 shows per-task results for offline RL evaluations. All results here are shown with a moving average over the last 5 evaluations.

We observe that despite the marked difference in log-likelihood-based preference learning evaluation between methods (Figure 1a), when the reward models produced by each method are used in the offline RL setting for policy optimization, they all led to policies of similar rollout performance. This lack of consistent correlation between supervised learning of reward models and reinforcement learning of policies is a known behavior in the RL from preference feedback literature, across both language modeling and control domains (Gao et al., 2022; Tien et al., 2022; Pan et al., 2021). We emphasize that the primary contribution of our work is an efficient reward learning algorithm for learning from preference feedback, and show that the resulting reward model can produce policies that reach competitive performance with policies that learn from reward models produced by other preference learning algorithms. We do not claim that our method can automatically lead to stronger policy performance. We leave investigation of the relationship between the learned reward posterior and how it affects policy optimization to further work (Razin et al., 2025; Swamy et al., 2025).

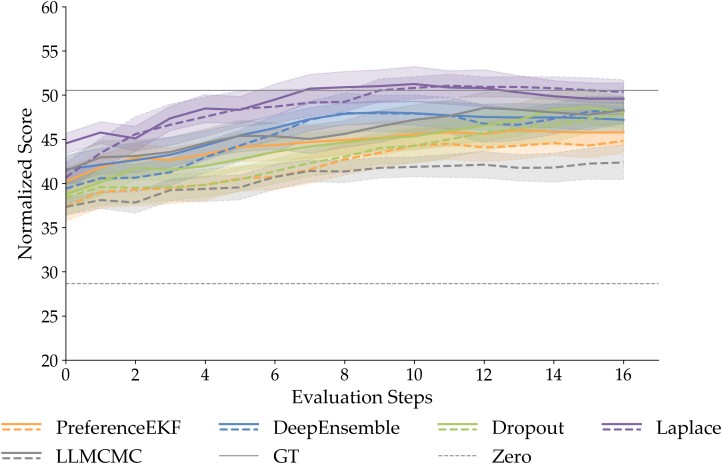

Figure A.14: Task-aggregate offline RL rollout performance across all tasks for each reward learning algorithm variant, all using infogain acquisition function: comparison of the random (dashed line) and active (solid line) variants of the learned reward models, with rollout performance aggregated over 12 D4RL tasks (mean±s.e. over 5 seeds).

### A.3.2   Implementation Details

The extent to which offline RL algorithms leverages reward information for policy optimization, i.e., whether reward-induced policy performance is a good metric for assessing learned reward models, is heavily dependent on the trajectory dataset: when ran on datasets consisting solely of expert demonstrations, offline RL algorithms will largely ignore reward information and adopt a behavioral cloning-like learning strategy. On the other hand, it is generally difficult to train a policy from a dataset consisting of purely random behavior (Kumar et al., 2021).

Following the experiment methodology of Shin et al. (2022) for our offline RL experiments, we add two reference performance scores to every task as shown in Figure A.15: we refer to "GT" as the score from an offline RL policy trained on $\mathcal{D}^{traj}$ labeled with ground-truth environment reward information, and "Zero" as score from a policy trained on $\mathcal{D}^{traj}$ with reward information zeroed out. This serves to test whether an offline RL algorithm is able to effectively leverage reward information for a given trajectory dataset. For most tasks, GT and Zero serve as upper and lower performance bounds for learned policies.

All offline RL experiments were done by running implicit Q-learning (IQL Kostrikov et al. (2021)) on trajectory transition datasets labeled with different types of rewards, e.g., ground truth environment reward, zeroed out reward, or preference-learned reward. An IQL agent consists of four neural networks: main and target

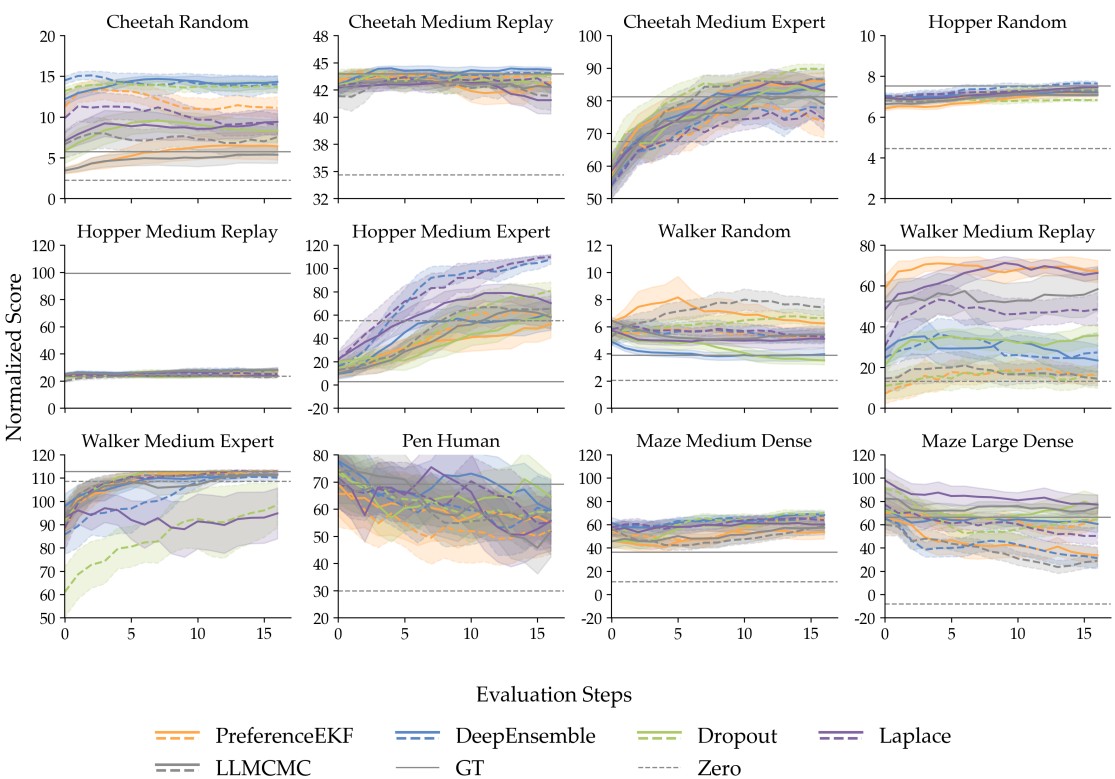

Figure A.15: Per-task offline RL rollout performance across all tasks for each reward learning algorithm variant, all using infogain acquisition function: comparison of the RM learned using random (dashed line) and active (solid line) variants of the algorithms across 12 D4RL tasks in the offline RL setting (mean±s.e. over 5 seeds). Black solid line indicates the performance of a policy trained on ground truth reward (GT), and black dotted line for a policy trained without reward information (Zero). In most tasks, active PreferenceEKF performs on par with other algorithms in terms of rollout score.

Q-network, a Gaussian policy network, and a state-value network. All four networks have two hidden layers of 256 units each and are trained using the same optimizer configuration with cosine decay learning rate schedule. Policy extraction is done with advantage-weighted regression (AWR Peng et al. (2019)). All training runs are done using 1M update steps with 5 rollouts every 50K steps for evaluation. We apply normalization to both reward and observation features, and further apply clipping for reward values exceeding 10. All hyperparameters are detailed in Table 4.

Table 4: Shared hyperparameters for IQL across all tasks. Here "Iterations" refers to the number of minibatch updates.

| Name | Value |
|------|-------|
| Optimizer | Adam |
| Learning rate | 0.0003 |
| Betas | (0.9, 0.999) |
| Iterations | 1M |
| Batch size | 256 |
| Discount factor $\gamma$ | 0.99 |
| Target net update step size | 0.005 |
| Expectile $\tau$ | 0.7 |
| Advantage temperature $\beta$ | 3.0 |
| Exponential advantage clip | 100 |

### A.4   Scaling Experiments.

JAX offers efficient function vectorization using `jax.vmap`. While we use this to parallelize ensemble model training and prediction in most experiments in Section 5, we do not use this for the scalability experiments in Section 5.2. Parallelized training and prediction of up to $M = 150$ models with up to 2M parameters (in the case of the three layer neural networks with 1024 units each) can lead to out-of-memory errors. We instead use python's native for loop to perform ensemble model training and prediction sequentially. All scalability experiments were done on CPU instead of GPU to avoid out-of-memory errors.

### A.5   LLM Usage

We used LLMs primarily for writing Python visualization scripts, figures/tables typesetting in Latex, finding related work on subspace construction methods, and debugging JAX compilation / model loading errors. We did not use LLMs for paper writing, research ideation, or implementing the core algorithm parts.

