# OpenReview forum: "Subspace Inference Enables Efficient Active Reward Learning from Preferences"
_TMLR — Under review for TMLR_

### Review · Reviewer_id39 · 2026-06-12

**Summary Of Contributions:**

The paper proposes PreferenceEKF, an active-learning Bayesian method that performs Kalman filtering in reduced-dimensionality subspace of neural networks to provide uncertainty estimates, from which multiple reward models can be sampled. Essentially, the idea is to scale InfoGain (acquisition function) to higher-dimensions, namely, neural networks, by using lower dimensional subspaces (learned).
The method is showcased on D4RL and V-D4RL, demonstrating better sample efficiency compared to other Bayesian approaches in reward learning, and performs comparably in preference-based offline RL settings.

**Additional Comments:**

**Strengths**:
* Rigorous experiments and evaluation (even though it is limited to synthetic datasets).
* I found the insights interesting overall.
* I appreciate the pre-trained image embeddings experiments.
* Detailed Appendix.
* Open-source code.
* Limitations are clearly discussed.


**Weaknesses**:
* Writing: I found the Abstract a bit vague – a lot of words around the general setting (RLHF, active learning), one vague sentence describing the method and then some insights. I would recommend spending more words about the actual method (e.g., a high-level description of “Bayesian filtering”, what are “neural network subspaces”, what does “maintain model posterior” mean and etc.). The method name (PreferenceEKF) should also appear in the abstract. The reader should get a general idea of what is the proposed method from the abstract.
* Motivation: I was missing some motivation for **why** we need this kind of approach for the tested scenarios (namely, simulated environments), why is this approach interesting compared to other reward-free settings?
* Synthetic data: the benchmark is focused on synthetic datasets, and it is unclear how it scales beyond the simulated environments. The preference labels are also synthetically generated (except for the crowdsourcing experiment, where the performance is not great). The observations are mostly ground-truth states (pixel-based has significant scalability issues due to the high-dimensions), which, again, limits this method. The pre-trained image embeddings is a nice touch, but again, limited to moderate dimensions.
* Dimensionality: the “high-dimensional neural networks” in question are 2-layer MLPs with 64 hidden units, which is arguably “deep”.


**Minor**:
* Page 2 - Related Work, penultimate paragraph: “UQ” is used without proper definition (I assume it is "Uncertainty Quantification”) .
Page 3 - final paragraph - similarly, RM appears without definition (I assume “Reward Model”).
* Figure 2b. - hard to to get a sense of the actual number due to the scale, maybe a Table would be better.
* All references to figures that appear in the Appendix, should explicitly mention that they are in the Appendix, as the reader should expect to jump to page 24 from page 8, for example.
* Page 10 - Section 5.5: “an reward model” - > “a reward model”.

**Audience:**

Yes

**Audience Explanation:**

Yes, the reinforcement learning community would be interested in the findings of this paper.

**Broader Impact Concerns:**

Clearly addressed.

**Claims And Evidence:**

Yes

**Claims Explanation:**

Yes, the claims are supported by clear evidence. See below for strengths.

**Requested Changes:**

* Question: why does it make sense to approximate the state dynamics as the identity function?
* Please address “Weaknesses” and “Minor”.

---

> ### Author Response · Authors · 2026-07-21
>
> We thank the reviewer for the kind words and thorough feedback.
>
> - “Why does it make sense to approximate the state dynamics as the identity function?”
> 	- While the general Bayes filtering formulation accounts for randomness in both the measurement and the system dynamics, when applied to reward learning, we assume that there is only one true unknown set of model parameters (which we treat as system state), and each new query is just another noisy measurement of it via the likelihood model $p(D \mid \theta)$. We deemed the identity function as appropriate for capturing the idea of “one true unknown set of parameters” that doesn’t inherently change over time.
> - Writing
> 	- We completely agree with this point and have updated the abstract accordingly to reflect the proposed method.
> - Motivation:
> 	- The sequential nature of PreferenceEKF allows it to learn from data (preference queries) as they arrive without the need to retrain on older data. All baseline methods require retraining on older data for posterior update so as to prevent catastrophic forgetting, but this increases runtime demand where our method has clear advantage (Figure 1b).
> 	- By taking advantage of the hypothesis that “the number of parameters required for a neural network to solve a task is often much smaller than the network’s total parameter count,” we believe that performing posterior inference in parameter subspace focuses the updates on parameters actually needed for solving the task, instead of redundant parameters commonly found in the overparameterized regime. This leads to our method having a clear sample efficiency advantage compared to other methods (Figure 1a.)
> 	- This combination of compute and sample efficiency is particularly important in active reward learning settings, where the primary application is human-in-the-loop learning. When receiving human preference feedback in real time, it is imperative that we 1) minimize the number of queries needed to learn the human’s preference, and 2) minimize the human’s wait time in-between queries as the model performs posterior update. We believe that the two aforementioned characteristics of our method make it a great fit for this problem setting.
> - Synthetic data
> 	- We added an additional experiment based on real-world robotics data collected by a fleet of robot arms across various manipulation tasks [1]. The trajectory observations are proprioceptive states encoding robot end effector state, and preference labels are generated over sampled pairs of successful and failed trajectories for each task.
> 	- Compared to the setting of our main results from section 5.1, where preference labels are generated synthetically from dense environment rewards, the sparse binary success feedback format of the newly included dataset is very common in the robot learning literature, where dense rewards are often difficult to obtain.
> 	- In section A.2.10, we see that PreferenceEKF and LLMCMC are, by far, two of the best performing methods in this setting. We believe this result successfully showcases the applicability of our method to more realistic control settings. We also added this result to Section 5.6.
> - Dimensionality:
> 	- While our main experiments are performed using two-layer MLPs, we dedicate the entirety of section 5.2 to study how our method scales to large networks and more model posterior samples.
> 	- We managed to scale up model parameter size up to 1M parameters. This is a significant improvement over prior state of the art in information-theoretic active reward learning in the control domain, which still use linear reward models [2] or Gaussian processes [3]. As far as we know, our work is among the first to extend the approach to neural network reward models by allowing scalable sampling of neural network parameter posteriors.
> 	- If the reviewer’s concern is not so much over model size, but over the word “deep” neural networks, then we agree with the sentiment that even at 1M parameters, 3 layer neural networks is not exactly deep. We would be happy to drop this description.
>
> Minor changes:
> - All references to appendix figures now are of the form “Figure A.x”, whereas main body figure references are of the form “Figure x”
> - “Figure 2b. - hard to to get a sense of the actual number...”
> 	- We provide the table form of both Figure 1b and Figure 2b in Section A.2.2. Note that for Figure 1b, we changed a hyperparameter setting of LLMCMC to reduce its runtime without decreasing its performance.
>
>
> [1] Z. Zhou, et al., “Autonomous Improvement of Instruction Following Skills via Foundation Models,” Oct. 15, 2024, _arXiv_: arXiv:2407.20635.
>
> [2] E. Bıyık, et al., “Asking Easy Questions: A User-Friendly Approach to Active Reward Learning,” in _Proceedings of the Conference on Robot Learning_, PMLR, May 2020
>
> [3] E. Bıyık, et al., “Active Preference-Based Gaussian Process Regression for Reward Learning and Optimization,” _The International Journal of Robotics Research_, vol. 43, no. 5

---

> > ### Comment · Reviewer_id39 · 2026-07-22
> > **Thank you for the clarifications**
> >
> > I thank the authors for their effort in addressing my concerns, which are resolved. I believe the paper is in good shape and can be accepted for publication.

---

### Review · Reviewer_gYRs · 2026-06-26

**Summary Of Contributions:**

The paper uses an extended Kalman filter (EKF) to maintain a posterior over reward models for an MDP in an active learning setting, where at each timestep the agent is allowed to query a single preference for a chosen pair of trajectories. In this setting, Bayesian methods allow for computing useful metrics in determining which pair to query such as InfoGain. To make the method computationally tractable, posterior inference is performed within a learned subspace constructed via SVD. A projection matrix then maps the parameters back to the full weight space to perform likelihood inference. The method is much more computationally efficient than other Bayesian deep learning approaches, and it obtains competitive performance in both reward modeling from preferences and in training control policies with the learned reward model.

Strengths
1. Both the subspace approach and the single-sample posterior update used by the RKF enable much cheaper posterior inference as compared to the baselines.
2. The authors demonstrate their method performs well relative to a reasonable, wide set of diverse baselines.
3. The authors provide a rigorous set of key ablations including different acquisition functions, an identical number of model samples for computing InfoGain, no active learning (random querying), no initial dataset for subspace construction/model initialization, and identical compute budgets.

Weaknesses
1. The authors note that it is unclear whether performing posterior inference in the subspace will be effective with larger models. Additionally, the authors also note the EKF learns a gaussian posterior which may have limited efficacy with crowd-sourced human preference datasets as it is unable to model multimodal distributions. These two factors potentially limit the utility of this approach for the most common application of learning from preferences: finetuning LLMs. However, I think this is only a minor weakness, as there exist other use cases for this approach in robotics, and popular applications are subject to change.
2. I have some concerns regarding the validity of the statistical tests used in section A.2.1 (explained in the following section).

**Audience:**

Yes

**Audience Explanation:**

Yes, I believe the effectiveness of subspace posterior inference and EKFs in this setting may be of interest to both RLHF practitioners (particularly in robotics or other applications where shallower networks are more common unlike fine-tuning LLMs) and those interested in using model uncertainty as an intrinsic reward for exploration when training policies in the traditional RL setting.

**Broader Impact Concerns:**

Sufficiently addressed.

**Claims And Evidence:**

No

**Claims Explanation:**

Overall the claims are supported, with a few key caveats. Below is an assessment of each of the main claims:

1. “Active reward learning using PreferenceEKF resulted in better sample efficiency (in the number of queries required) compared to learning from random queries.”
I think this claim is potentially not well supported. Figure 1a is meant to support this claim, but the orange dashed and solid lines are quite close and the error bars are standard errors calculated over 5 seeds. Non-overlapping standard error bars do not indicate statistically significant differences (they are substantially narrower than confidence intervals at small sample sizes) [1]. Additionally, Table 1 is meant to support this claim, however at n=5 the paired t-test's p-values heavily depend on the difference in scores being Gaussian, and it’s not possible to check normality from five points [3]. Is this test appropriate for supporting this claim?

2. “PreferenceEKF performs on par with or better than all Bayesian deep learning baselines in terms of sample efficiency and calibration in preference modeling tasks.”
I believe this claim is supported by Fig 1a, though the on-par qualification is key and the same concerns from (1) above remain regarding the statistical tests and error bars, especially when seeking to claim PreferenceEKF performs better than the other methods.

3. “PreferenceEKF’s runtime is not only much faster, but also scales better with both model size and  number of posterior samples compared to all other methods.”
Supported by figure 2

4. “When used for policy optimization, reward models learned using PreferenceEKF resulted in policies with performance competitive with those optimized with reward models learned using other methods.”
Supported by Figure A.12 and A.13

**Requested Changes:**

Critical Changes
1. If the issues with the statistical tests in Table 1 outlined above hold, then I think more seeds should be run such that the total number of seeds is 10+, or the claim that active reward learning “resulted in better sample efficiency” than random queries with PreferenceEKF should be relaxed. If total seeds can be increased, then replacing standard error and standard deviation with bootstrapped confidence intervals would be more appropriate for the plots, since standard error and standard deviation do not support the significance comparisons.
2. I believe the definition of the return on the second line of 5.1 is incorrect. $R_\theta(\tau_i) = \frac{1}{T} \sum_{t=1}^{T} r_\theta(s_{i,t})$ is the average reward over a trajectory, while return is defined as the sum of the rewards $ \sum_{t=1}^{T} r_\theta(s_{i,t})$. If the environments used in the analysis always have episodes of fixed length, then this does not impact the analysis, otherwise I believe the analyses in A.3.1 should be rerun with the proper return calculation.

Non-critical changes
1. For the plots that compare mean normalized performance across several environments, a detailed description of how the per-environment normalization is performed would be beneficial for clarity.
2. “We use nlinearize = 5, prior noise of 0.07, systems noise of 1e-3, and measurement noise of 0.07 for all of our PreferenceEKF experiments.” An explanation on if these hyperparameters were tuned and if yes, how they were tuned, would be beneficial.
3. There are no training curves for baseline methods reported in Figures A.9 and A.10, which makes interpreting the quality of the performance difficult.

Typos
1. “popular uncertainty representations” → “popular uncertainty representation”
2. “Predictive distribution for computing InfoGain” → “The predictive distribution…”
3. “PreferenceEKF learn from only the” → “PreferenceEKF learns from only the”

---

> ### Author Response · Authors · 2026-07-21
>
> We thank the reviewer for the kind words and thorough feedback. Based on their comments, we made three major changes in the revised version as we elaborate below.
>
> (1) For our main preference learning results in Sections 5.1 and 5.2, we re-ran the experiments with 12 seeds instead of 5, and replaced the standard error from Figure 1 with 95% bootstrapped confidence intervals. As the reviewer hinted, we indeed observed that the new error bars are broader compared to the ones constructed using standard error. We also conducted hypothesis testing in Section A.2.1 using bootstrapped resampling to re-evaluate the following claims: (a) “Active reward learning using PreferenceEKF resulted in better sample efficiency (in the number of queries required) compared to learning from random queries.” and (b) “PreferenceEKF performs on par with or better than all Bayesian deep learning baselines in terms of sample efficiency and calibration in preference modeling tasks.”
>
> Under the new statistical tests and sample size, we document in Table 1 that the active variant of PreferenceEKF no longer outperforms its random counterpart with statistical significance; we thus no longer make the first claim in our paper, but still emphasize that both active and random PreferenceEKF are among the best performing methods across all tasks. We also see that PreferenceEKF outperforms all baselines except for LLMCMC with statistical significance; since our method performs on par with LLMCMC as seen in Figure 1, the second claim remains valid. We thank the reviewer again in helping us correct and strengthen our claims.
>
> (2) We thank the reviewer for catching the critical typo in our formulation. The return definition of our policy learning experiments is indeed the sum of rewards without normalization by trajectory length. The analyses in A.3.1 remain valid the way they currently are. This typo comes from the fact that when we produce synthetic preference labels for our preference learning experiments (independent of policy learning), we do use length-normalized returns for numerical stability under the Bradley-Terry model. But this does not change the generated labels since we use fixed-length trajectories for preference comparison, making this a minor implementation detail.
>
> (3) “However, I think this is only a minor weakness, as there exist other use cases for this approach in robotics, and popular applications are subject to change.”
>
> We added an additional experiment based on real-world robotics data collected by a fleet of robot arms across various manipulation tasks [1]. Compared to the setting of our main results from section 5.1, where preference labels are generated synthetically from dense environment rewards, we leverage sparse binary success labels to generate preference feedback. We note that this sparse trajectory data quality signal is very common in the robot learning literature, where dense rewards are often difficult to obtain.
>
> In section A.2.10, we see that PreferenceEKF and LLMCMC are, by far, two of the best performing methods for preference learning in this new setting. We believe this result successfully showcases the applicability of our method to more realistic control settings. We also added this result to main paper Section 5.6.
>
> Additionally, we made the following minor changes in the paper based on the reviewer’s comments:
> 1. We detail the per-environment normalization scheme in the second paragraph of the Appendix on page 19.
> 2. We provided further explanations of the PreferenceEKF hyperparameters and their tuning in Appendix A.1.3.
> 3. For pixel-based reward modeling (Figures A.11 & A.12 in the revised paper), we provided additional results on two of the four baseline methods: DeepEnsemble and Dropout. We excluded Laplace approximation and LLMCMC as they took too long to run on the ResNet-based pixel reward model. Among the 3 pixel-based task datasets, PreferenceEKF has the best final test log-likelihood among the evaluated methods, with its task-aggregate performance being slightly better than DeepEnsemble.
> 4. All typos have been addressed in the revised paper. We thank the reviewer for catching them!

---

> ### Comment · Reviewer_gYRs · 2026-07-22
>
> Looks great, thank you for taking the time to make these thorough and thoughtful revisions! Just one minor note:
> In the text of Section A.2.1, it mentions that Laplace and LLMCMC failed to outperform their random counterparts, but Table 1 shows p = 0.011 and p < 0.001 for these two methods, respectively. I am wondering if this is a typo after the change in tests such that they are actually no longer significant, or perhaps the text is outdated?

---

> > ### Author Response · Authors · 2026-07-22
> >
> > We thank the reviewer for catching this error! Yes, it was outdated text. We have uploaded a new version that fixes this mistake.
> >
> > We also left out a citation in our first reply to your review, under
> >
> > > (3) “However, I think this is only a minor weakness, as there exist other use cases for this approach in robotics, and popular applications are subject to change.”
> >
> > To which we replied
> > > We added an additional experiment based on real-world robotics data collected by a fleet of robot arms across various manipulation tasks [1]....
> >
> > The missing citation is as follows:
> >
> > [1] Z. Zhou, et al., “Autonomous Improvement of Instruction Following Skills via Foundation Models,” Oct. 15, 2024, _arXiv_: arXiv:2407.20635.

---

### Review · Reviewer_QrrS · 2026-07-07

**Summary Of Contributions:**

Active learning can help to address the problem of poor sample efficiency in reinforcement learning from human feedback. But uncertainty quantification remains challenging. The paper develops a method that leverages Bayesian filtering in neural network subspaces to efficiently maintain model posterior for active reward modeling in continuous control tasks. This proposed approach enables scalable sampling of neural network reward models to efficiently compute active learning acquisition functions. Numerical experiments demonstrate the efficiency of the proposed method.

**Audience:**

Yes

**Audience Explanation:**

I think so. This topic itself should be of interest to anyone who is interested in reinforcement learning from human feedback, active learning and Bayesian methods.

**Claims And Evidence:**

Yes

**Claims Explanation:**

I think so. There are extensive numerical studies to support the claims of the findings summarized in the last part in Section 1 in the paper.

**Requested Changes:**

(1) In my view, the biggest weakness of the paper is lack of explanations of intuitions why the proposed method works well. For example,
In Section 5.1., the paper gives a positive answer to the question whether PreferenceEKF leads to sample-efficient active reward learning, but it is only through numerical experiments. I understand this is not a theory paper, but it would be really helpful if the author(s) can add some discussions why the answer to this question is positive and what is the intuition behind the effectiveness of the proposed method. Similarly, in Section 5.2., the paper shows that the training runtime scales well, but again, there is a lack of discussions on the intuition why the training runtime of the proposed method scales well. Similarly, in Section 5.3, the paper shows better model calibration. Again, more discussions on why it leads to better model calibration will be very helpful.

(2) For equation (3), please combine the two lines into one line.

(3) On page 7, it is better to write Figure 1a instead of Fig. 1a because  for Section, you do not use the abbreviation Sec. It is better to be consistent.

(4) In the first paragraph in Section 5.1., Appendix Section A.2.1 is redundant. It should be either Appendix A.2.1 or Section A.2.1. The same can be said about Appendix Section A.2.6. in the last paragraph  on page 7, Appendix Section A.2.10 in the last paragraph in Section 5.3, and Appendix Section A.2.7 in the second paragraph in Section 5.4.

(5) The comma right after equation (7) should be put inside equation (7). The same can be said about equation (8) and the displayed equation after equation (8).

(6) On page 2, --it been shown to lead to-- is a typo.

(7) On page 5, See A.2.2 should be See Section A.2.2 or Appendix A.2.2.

(8) Please explain how you get (2b) from (2a) or at least provide a reference. Please also provide a brief explanation or provide reference that as $M\rightarrow\infty$ in equation (3), you get equation (2).

---

> ### Author Response · Authors · 2026-07-21
>
> We thank the reviewer for the kind words and helpful feedback, and will go through the requested changes one by one:
>
> - Change (1)
> 	- We appreciate the reviewer’s introspective questions that allow us to explain some of the mechanism behind our approach!
> 	- For Section 5.1’s sample efficiency result:
> 		- By taking advantage of the hypothesis that “the number of parameters required for a neural network to solve a task is often much smaller than the network’s total parameter count,” we believe that performing posterior inference in parameter subspace focuses the updates on parameters actually needed for solving the task, instead of redundant parameters commonly found in the overparameterized regime. This leads to our method having a clear sample efficiency advantage compared to other methods (Figure 1a.).
> 		- Among the baselines, LLMCMC is the only other method that leverages this hypothesis by performing MCMC over only the final layer of a reward model. As we see in Figure 1a., its performance is on par with PreferenceEKF, but its runtime is dramatically slower compared to all methods due to the sequential nature of MCMC (Figure 1b.).
> 		- We further note the distinction between parameter subspace vs. parameter subset. While PreferenceEKF uses a projected subspace, LLMCMC uses a subset of the full parameter count within the last layer. While the question of which approach can find more task-critical parameters is out of the scope of our work, we find it an interesting point to bring to the discussion.
> 	- For Section 5.2’s runtime result:
> 		- The sequential nature of PreferenceEKF allows it to learn from data (preference queries) as they arrive without the need to retrain on older data. All baseline methods require retraining on older data over multiple minibatches for posterior update so as to prevent catastrophic forgetting, but this increases runtime demand. In contrast, our method only needs to perform EKF update step on the most recently received query, giving it a clear runtime advantage (Figure 1b). We further note that PreferenceEKF is the only evaluated method in our work to exhibit this property.
> 		- Convergence of MCMC chains can be slow, which explains why MCMC is by far the slowest method (Figure 1b) despite its excellent log-likelihood performance (Figure 1a). Laplace baseline requires SGD in addition to Laplace approximation on every new query, making it the second slowest method.
> 	- For Section 5.3’s calibration result
> 		- While Brier score measures both model confidence and calibration, ECE more directly measures calibration, and will be the focus of the following points. (See top plot of Figure 3a.)
> 		- PreferenceEKF maintains an explicit Gaussian posterior over subspace parameters (mean and full covariance), updated sequentially via EKF using Bradley–Terry likelihood as measurement model. Predictive distributions are computed by averaging preference predictions over posterior samples. The calibration of a prediction is therefore tied directly to epistemic uncertainty over subspace parameters, rather than to optimization artifacts such as parameter initializations and dropout masks that the baselines DeepEnsemble and Dropout rely on.
> 		- While optimization artifacts don’t always work for posterior inference, they can still be somewhat effective approximation tools, as shown by DeepEnsemble’s middling but not terrible performance in Figure 3a.
> 		- Laplace approximates the posterior locally around a MAP point obtained from SGD on all observed queries; for neural reward models, this local Gaussian approximation around a MAP point can often be miscalibrated, resulting in it having some of the worse ECE results in Figure 3a.
> 		- LLMCMC is the most principled baseline, and indeed its ECE result in Figure 3a is closer to PreferenceEKF. But it is restricted to a subset of parameters in the final layer, as opposed to PreferenceEKF’s subspace within the full parameter space, which may explain why it underperforms PreferenceEKF.
> - Changes (2) - (7)
> 	- We thank the reviewer for the suggestions which, we believe, improve the readability of our paper. We have incorporated all the suggestions in the revised version of the paper.
> - Change (8)
> 	- In the revised paper, we provide a derivation of Equation 2b from Equation 2a in appendix section A.2.5 based on the definition of mutual information. For the derivation of Equation 3, the sampling-based approximation of the InfoGain objective from Equation 2b, we added a sentence in the main text to refer to [1] which we also detail in section A.2.5.
>
> [1] E. Bıyık, M. Palan, N. C. Landolfi, D. P. Losey, and D. Sadigh, “Asking Easy Questions: A User-Friendly Approach to Active Reward Learning,” in _Proceedings of the Conference on Robot Learning_, PMLR, May 2020, pp. 1177–1190.